# Aedes mosquitoes acquire and transmit Zika virus by breeding in contaminated aquatic environments

Senyan Du[1,2,3], Yang Liu[1,4], Jianying Liu[1], Jie Zhao[1], Clara Champagne[5], Liangqin Tong[1], Renli Zhang[3], Fuchun Zhang[6], Cheng-Feng Qin [7], Ping Ma[8], Chun-Hong Chen[9], Guodong Liang[10], Qiyong Liu[11], Pei-Yong Shi[12], Bernard Cazelles [5,13], Penghua Wang[14], Huaiyu Tian[15] & Gong Cheng[1,2,3]

Zika virus (ZIKV) is a mosquito-borne flavivirus that predominantly circulates between humans and *Aedes* mosquitoes. Clinical studies have shown that Zika viruria in patients persists for an extended period, and results in infectious virions being excreted. Here, we demonstrate that *Aedes* mosquitoes are permissive to ZIKV infection when breeding in urine or sewage containing low concentrations of ZIKV. Mosquito larvae and pupae, including from field *Aedes aegypti* can acquire ZIKV from contaminated aquatic systems, resulting in ZIKV infection of adult females. Adult mosquitoes can transmit infectious virions to susceptible type I/II interferon receptor-deficient (*ifnagr-/-*) C57BL/6 (AG6) mice. Furthermore, ZIKV viruria from infected AG6 mice can causes mosquito infection during the aquatic life stages. Our studies suggest that infectious urine could be a natural ZIKV source, which is potentially transmissible to mosquitoes when breeding in an aquatic environment.

[1] Tsinghua-Peking Joint Center for Life Sciences, School of Medicine, Tsinghua University, 100084 Beijing, People's Republic of China. [2] Beijing Advanced Innovation Center for Structural Biology, School of Medicine, Tsinghua University, 100084 Beijing, People's Republic of China. [3] Institute of Pathogenic Organisms, Shenzhen Center for Disease Control and Prevention, 518055 Shenzhen, Guangdong, People's Republic of China. [4] School of Life Science, Tsinghua University, 100084 Beijing, People's Republic of China. [5] IBENS, UMR 8197 CNRS-ENS Ecole Normale Supérieure, 75005, Paris, France. [6] Institute of infectious diseases, Guangzhou Eighth People's Hospital, Guangzhou Medical University, 510060 Guangzhou, People's Republic of China. [7] State Key Laboratory of Pathogen and Biosecurity, Beijing Institute of Microbiology and Epidemiology, 100071 Beijing, People's Republic of China. [8] Department of Clinical Laboratory, Tsinghua University Hospital, 100084 Beijing, People's Republic of China. [9] National Institute of Infectious Diseases and Vaccinology, National Health Research Institutes, Zhunan, Miaoli, 35053 Taiwan, Republic of China. [10] State Key Laboratory for Infectious Disease Prevention and Control, Collaborative Innovation Center for Diagnosis and Treatment of Infectious Diseases, National Institute for Viral Disease Control and Prevention, China CDC, 102206 Beijing, People's Republic of China. [11] State Key Laboratory of Infectious Disease Prevention and Control, Collaborative Innovation Center for Diagnosis and Treatment of Infectious Diseases, National Institute for Communicable Disease Control and Prevention, China CDC, 102206 Beijing, People's Republic of China. [12] Department of Biochemistry and Molecular Biology, Department of Pharmacology and Toxicology, and Sealy Center for Structural Biology and Molecular Biophysics, University of Texas Medical Branch, Galveston, TX 77555, USA. [13] International Center for Mathematical and Computational Modeling of Complex Systems (UMMISCO), IRD-Sorbone Université, Bondy 93143, France. [14] Department of Immunology, School of Medicine, the University of Connecticut Health Center, Farmington, CT 06030, USA. [15] State Key Laboratory of Remote Sensing Science, College of Global Change and Earth System Science, Beijing Normal University, 100875 Beijing, People's Republic of China. These authors contributed equally: Senyan Du, Yang Liu, Jianying Liu. Correspondence and requests for materials should be addressed to H.T. (email: tianhuaiyu@gmail.com) or to G.C. (email: gongcheng@mail.tsinghua.edu.cn)

Zika virus (ZIKV) is a mosquito-borne virus belonging to the genus *Flavivirus* and is transmitted to humans by mosquitoes of the genus *Aedes*. Both *Aedes aegypti* and *Aedes albopictus* are the primary vectors for ZIKV transmission in nature[1,2]. Recent ZIKV epidemics in the Americas resulted in more than 223,000 confirmed cases until the end of 2017[3]. Several neurological complications, such as Guillain-Barré syndrome in adults[4], and microcephaly in neonates[5], are associated with ZIKV infection. As a mosquito-borne flavivirus, ZIKV is well known to survive in a transmission cycle between mosquitoes and humans[6]. After transmission by infected mosquitoes feeding on naive hosts, ZIKV rapidly develops viremia, which enables acquisition by other mosquitoes via a blood meal. The virus subsequently infects the midgut epithelial cells and systemically invades other tissues, such as the salivary glands. Consequently, the infected mosquitoes transmit the virus to another host through blood feeding. In addition to causing viremia in host blood, ZIKV has also been isolated from host semen, urine, saliva, amniotic fluid, breast milk, and cerebrospinal fluid[7–10]. Transmission of ZIKV has also been suggested to occur via other nonvector approaches, such as sexual, congenital transmission, and blood transfusions[11–13].

Accumulating evidence indicates that infectious ZIKV particles are excreted into urine[14–16]. To date, at least five ZIKV infectious strains have been recovered and cultured from human urine samples[14,16–19], demonstrating that infectious ZIKV is discharged by patient viruria. Another study reports an infectious ZIKV titer of 10 pfu/ml in the urine of a patient[14]. In addition, ZIKV viruria may persist for several days. The equivalent of 12–20 pfu/ml ZIKV was present in urine samples from three patients from 5 to 26 days after the onset of Zika symptoms[20]. Indeed, an adult may urinate 1000–2000 ml daily and, if infected by ZIKV, discharge infectious ZIKV into the environment, including sewers, water pools, septic tanks, or other artificial containers around human dwellings. In a disease-outbreak scenario, many infected individuals (with or without symptoms) may intensively discharge a large volume of infectious urine into a restricted aquatic habitat of mosquitoes. We therefore investigated whether the urine from Zika patients could be a ZIKV source that facilitates mosquito infection in aquatic stages, and thus enables viral transmission.

In this study, we demonstrate that both *A. aegypti* and *A. albopictus* are permissive to ZIKV infection when breeding in urine or sewage containing low concentrations of ZIKV. ZIKV viruria from infected AG6 mice leads to mosquito infection during breeding. Furthermore, female field *A. aegypti* breeding in infectious sewage can transmit infectious Zika virions to naive AG6 mice, suggesting that ZIKV in contaminated aquatic environments is transmissible by mosquitoes during breeding.

## Results

**Mosquitoes are permissive to ZIKV infection when breeding**. Both *A. aegypti* and *A. albopictus*, which live around human dwellings, are major vectors for ZIKV transmission[21]. These *Aedes* species commonly breed in containers with clean water such as tree holes, flower pots, and tires. However, accumulating evidence from field surveillance convincingly shows that *A. aegypti* and *A. albopictus* tend to oviposit and breed in wastewater with low dissolved oxygen and high turbidity, such as cesspits, septic tanks, and sewers (Supplementary Table 1). Notably, immature forms of *A. aegypti* have been widely found in raw sewage during dengue epidemics in Brazil[22]. The population of adult *A. aegypti* mosquitoes emerging from wastewater may be equal or even larger than those from traditional breeding sites (clean water) in Brazil[23] and Puerto Rico[24]. We therefore wondered whether mosquitoes might acquire ZIKV while breeding in

aquatic conditions with human urine containing infectious ZIKV. We first assessed ZIKV stability in human urine. To mimic the conditions in the human bladder, we incubated ZIKV with eight fresh human urine samples (PRVABC59 strain, final titer was $1 \times 10^5$ pfu/ml) and incubated ZIKV with phosphate-buffered saline (PBS) as a control. The mixture was then maintained at 37 °C (human physiological temperature) for 48 h. Interestingly, ZIKV survivability varied significantly between individual urine samples. A titer of ZIKV ($>10^2$ pfu/ml) was detected in two samples at 24 h at 37 °C (Fig. 1a). To mimic the natural post-excretion conditions, we incubated ZIKV with the same eight fresh human urine samples (PRVABC59 strain, final titer was 20 pfu/ml). Incubation of ZIKV with PBS served as a control. The mixture was then maintained at 28 °C for 48 h. A titer of ZIKV was detected in one sample at 24 h at 28 °C using a plaque assay (Fig. 1b). This result suggests that the extended stability of ZIKV in human urine in the environment may provide a time window for ZIKV infection during mosquito breeding.

We next investigated whether the mosquitoes that breed in infectious urine might acquire ZIKV. We first assessed the susceptibility of larval and pupal stages to ZIKV infection in aquatic conditions. The fresh urine from Donor 3 (Fig. 1a) was mixed with ZIKV supernatant from infected Vero cells for *A. aegypti* larvae and pupae breeding (PRVABC59 strain, final titer was 20 pfu/ml). After exposure to ZIKV urine, the larvae or pupae were transferred to a new container with fresh water until eclosion (Supplementary Fig. 1a). The emerging adult mosquitoes were reared for 8 days for ZIKV detection by quantitative reverse transcription PCR (RT-qPCR). Intriguingly, both the larvae and pupae in the ZIKV urine were permissive to ZIKV infection and carried ZIKV throughout development to adults (Supplementary Fig. 1b). Notably, the mosquitoes emerging from pupae incubation presented the highest prevalence of ZIKV infection, suggesting that the pupae might be more susceptible to ZIKV in urine.

Both *A. aegypti* and *A. albopictus* have been shown to carry and transmit ZIKV in several field surveillance reports[1,2]. We next investigated whether the different mosquito species breeding in human urine with ZIKV might acquire an infection. In the first experiment, we incubated the pupae of either *A. aegypti* (Rockefeller strain) or *A. albopictus* (Jiangsu strain) with the ZIKV-positive urine, in which the final ZIKV titers were 20, 2, and 0.2 pfu/ml (Fig. 1c). The pupae of the two mosquito species emerged in human urine (Donor 3) (Supplementary Fig. 2a). The adult mosquitoes were reared for 8 days and subjected to ZIKV detection by RT-qPCR. For the *A. aegypti* Rockefeller strain, 3.7% (8/215) and 2.7% (4/147) of mosquitoes in the adult stage, breeding in 20 and 2 pfu/ml ZIKV urine, respectively, were positive for ZIKV by RT-qPCR detection. However, none of the mosquitoes (0/129) acquired the infection from human urine with 0.2 pfu/ml ZIKV (Fig. 1d). Additionally, 2.7% (6/223) and 0.7% (1/149) of adult *A. albopictus* mosquitoes breeding in 20 and 2 pfu/ml ZIKV urine, respectively, showed positive detection of ZIKV (Fig. 1e). We next tested the infectivity of ZIKV virions in mosquitoes breeding in ZIKV urine. In human urine with 20 pfu/ml ZIKV, 2 of 50 *A. aegypti* pools (10 mosquitoes per pool) and 1 of 93 *A. albopictus* pools were positive for infectious ZIKV, as determined by ZIKV culture in Vero cells. However, under similar experimental conditions with 2 pfu/ml ZIKV, 1 of 65 *A. aegypti* pools was positive for the infectious virions, while no infection was detected from a total of 100 pools of *A. albopictus* (Table 1). The infectious supernatant was further subjected to viral sequencing to confirm that the cytopathic effects were caused by ZIKV infection. Thus, we demonstrate that different *Aedes* mosquito species are susceptible to ZIKV infection when breeding in infectious aquatic systems.

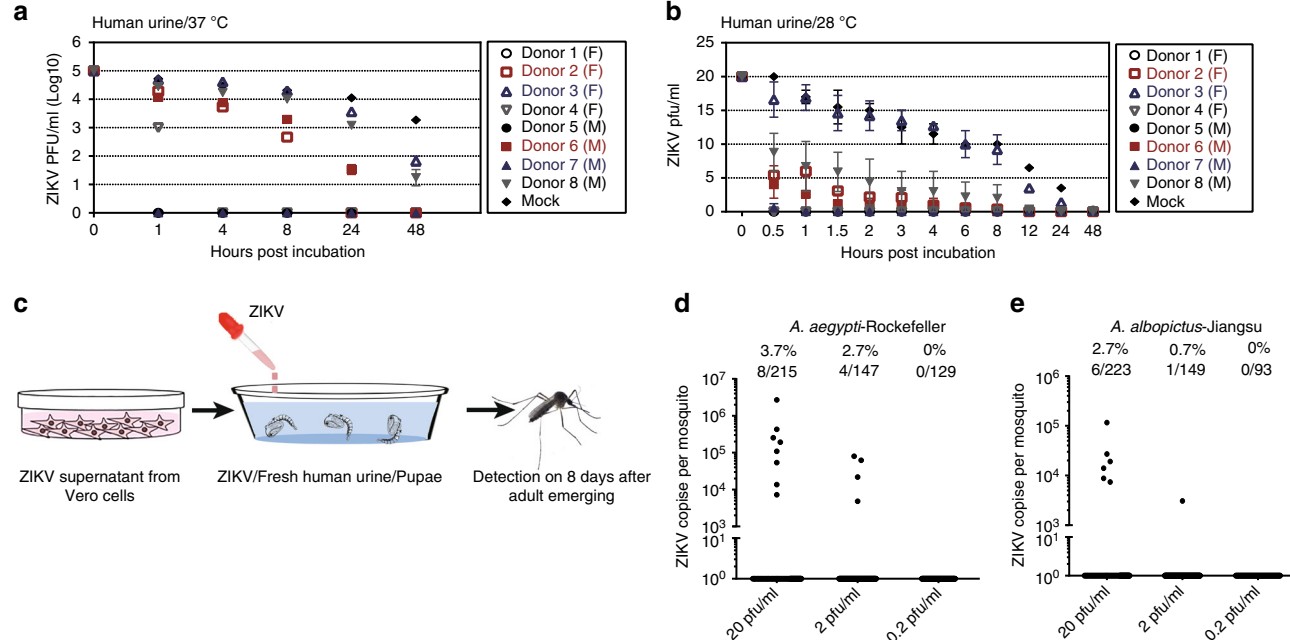

**Fig. 1** Mosquitoes are permissive to Zika virus (ZIKV) infection during breeding. **a**, **b** ZIKV survivability in human urine. ZIKV was incubated with fresh human urine or phosphate-buffered saline (PBS) (Mock). The mixtures were then maintained at either 37 °C or 28 °C over a time course and subsequently subjected to a viral titration assay. The PRVABC59 strain was used for the incubation. The initial ZIKV titer was $1 \times 10^5$ or 20 pfu/ml. Data are presented as the mean ± S.E.M. **a** $n = 3$ biologically independent samples. **b** $n = 2$ biologically independent samples. The data were repeated by two independent experiments with the similar results. F: female; M: male. **c–e** Acquisition of ZIKV infection by *Aedes* mosquitoes breeding in infectious human urine. **c** Experimental schematic representation. Freshly emerging mosquito pupae were used for breeding in ZIKV urine. The urine from human Donor 3 (Fig. 1a) was used in these experiments. Both ZIKV prevalence and infectivity were determined in the *A. aegypti* Rockefeller strain (**d**) and the *A. albopictus* Jiangsu strain (**e**). The mosquitoes were breeding from human urine with serial ZIKV titration. The emerging adults were reared for an additional 8 days for ZIKV detection by TaqMan quantitative PCR (qPCR). The number of infected mosquitoes relative to the total number of mosquitoes is shown at the top of each column. One dot represents one mosquito. The percentages are represented as the ratios of mosquito infection. **a–e** Source data are provided in as Source Data file

| Table 1 Measurement of infectious ZIKV particles in mosquitoes breeding in ZIKV urine | | | |
|---|---|---|---|
| **Virus titer** | **Mosquito species** | **Number of mosquito pools**[a] | **Positive pools for infectious virions** |
| 20 pfu/ml | *Aedes aegypti* -Rockefeller | 50 | 2 |
| | *Aedes albopictus* -Jiangsu | 93 | 1 |
| 2 pfu/ml | *Aedes aegypti* -Rockefeller | 65 | 1 |
| | *Aedes albopictus* -Jiangsu | 100 | 0 |

*ZIKV* Zika virus, *pfu* plaque-forming unit
[a]Ten mosquitoes per pool

We next assessed whether mosquitoes breeding in ZIKV urine could transmit ZIKV to a naive host. A mosquito-mouse-mosquito transmission model has previously been used for a ZIKV transmission study[25]. We first reared the pupae of the *A. aegypti* Rockefeller strain in urine with 20 pfu/ml ZIKV. Eight days after the adults emerged, twenty female *A. aegypti* mosquitoes were allowed to feed on a type I/II interferon (IFN) receptor-deficient (*ifnagr*−/−) C57BL/6 (AG6) mouse, which is an established animal model for ZIKV infection[25] (Fig. 2a). Notably, 3 out of the 17 mice exposed to infected mosquitoes were ZIKV-positive by RT-qPCR detection at least one day post infection (Fig. 2b). The infectious ZIKV particles were also detected at the viremic peak by a plaque assay (Fig. 2c). We next validated this phenomenon with a field-derived *A. aegypti* strain, which was collected from the ZIKV epidemic area of Paraiba, Brazil. RT-qPCR revealed that 2.3% (6/258) and 1.6% (3/191) of emerged adult mosquitoes (Brazil Paraiba strain), which bred in either 20 or 2 pfu/ml ZIKV urine, respectively, were positive for

ZIKV (Fig. 2d). Subsequently, the emerged female *A. aegypti* mosquitoes were fed on the AG6 mice with the same experimental procedure as used for the Rockefeller strain (Fig. 2a). One out of 14 mice exposed to infected mosquitoes developed robust viremia from 1 to 7 days post feeding (Fig. 2e, f). This infected mouse died at 10 days post feeding, indicating efficient ZIKV transmission by the mosquitoes breeding in the ZIKV urine.

We next assessed this transmission route for other arboviruses. In contrast to ZIKV, Dengue virus 2 (DENV2) rapidly lost its infectivity in fresh human urine at 37 °C (Supplementary Fig. 3a). A previous study reported the existence of DENV2 RNA in urine discharged from a patient with dengue fever[26]. Intriguingly, 2 of 109 *A. aegypti* mosquitoes (the Rockefeller strain) acquired DENV2 infection during pupae breeding (Supplementary Fig. 3b). We tried to isolate infective DENV virions from mosquitoes breeding in urine containing DENV2 using the same experimental procedure as in Table 1. However, no positive samples were identified from 50 mosquito pools (10 mosquitoes per pool).

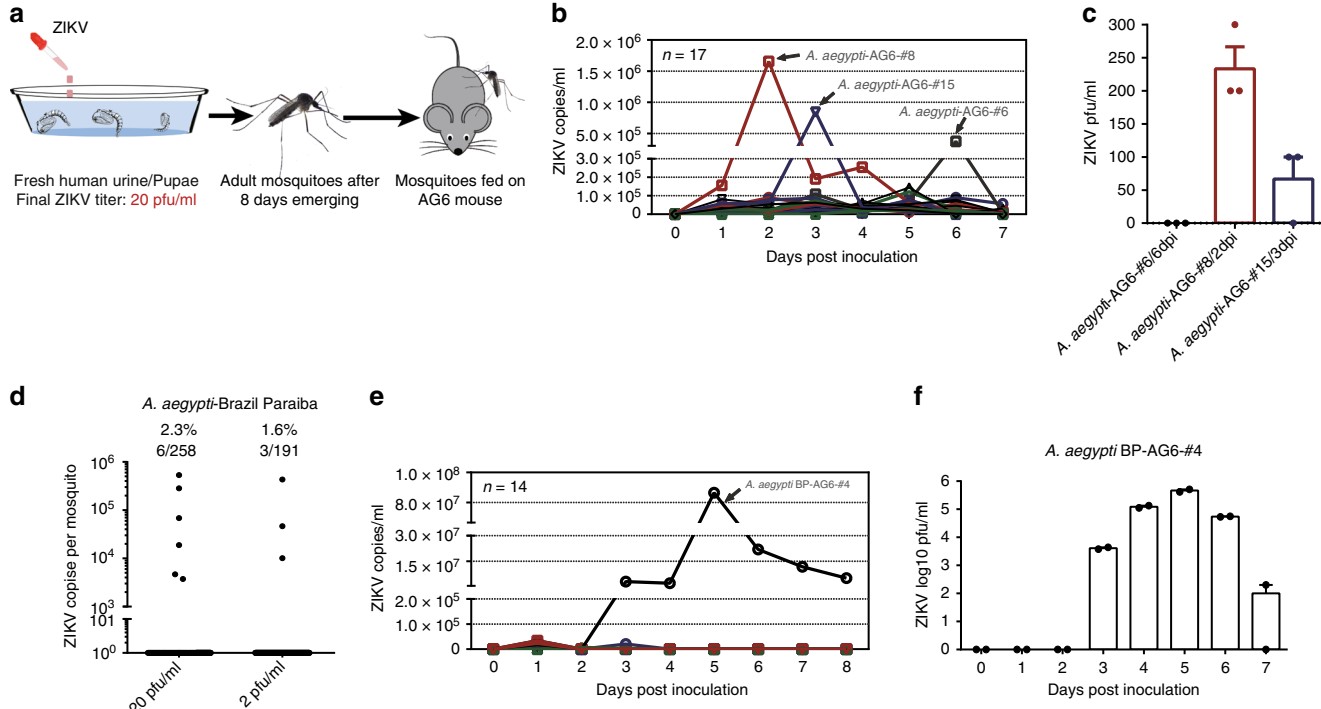

Fig. 2 Zika virus (ZIKV) is transmissible between AG6 mice and mosquitoes breeding in ZIKV urine. **a** Schematic representation of the ZIKV transmission between AG6 mice and mosquitoes breeding in the ZIKV urine. The *A. aegypti* pupae were breeding in the ZIKV urine (final ZIKV titer was 20 pfu/ml). The urine from human Donor 3 (Fig. 1a) was used in these experiments. After rearing for 8 days after adults emerged, 20 emerged female mosquitoes were allowed to feed together on an AG6 mouse. The mouse viremia was determined over a time course. **b**, **c** ZIKV viremia in the mosquito-bitten AG6 mice. The *A. aegypti* Rockefeller strain was used in this experiment. **d** Acquisition of ZIKV infection by a field *A. aegypti* Brazil Paraiba strain breeding in infectious human urine. With the same experimental procedure as Fig. 1c, both ZIKV prevalence and infectivity were determined in the *A. aegypti* Brazil Paraiba strain breeding in human urine with a serial ZIKV titration. The emerging adults were reared for 8 days for subsequent ZIKV detection by TaqMan quantitative PCR (qPCR). The number of infected mosquitoes relative to the total number of mosquitoes is shown at the top of each column. One dot represents one mosquito. The percentages are represented as the ratios of mosquito infection. **e**, **f** ZIKV viremia in the AG6 mice bitten by the *A. aegypti* Brazil Paraiba mosquitoes breeding in human urine. **b**, **e** The presence of ZIKV RNA copies in whole blood was assessed using quantitative reverse transcription PCR (RT-qPCR) detection. *n* represents the mouse number used in the experiment. The results were pooled from three detection replicates. **c**, **f** The numbers of infectious particles at the viremic peak were detected by a plaque assay. **c** $n = 3$ detection replicates. **f** $n = 2$ detection replicates. Data are presented as the mean ± SEM. **b**–**f** Source data are provided as a Source Data file

In addition, both the Batai virus (BATV, *Orthobunyavirus*) and Sindbis virus (SINV, *Alphavirus*) showed stable infectivity after 8 h of incubation at 37 °C (Supplementary Fig. 3c, e). However, the *Culex quinquefasciatus* mosquitoes, which emerged from pupae, failed to acquire BATV or SINV from the urine (Supplementary Fig. 3d, f).

Clinical surveillance has shown a relatively low ZIKV viremia ($10^2$–$10^3$ pfu/ml) in serum samples[8,27,28], compared to that of Chikungunya virus (CHIKV) and Dengue virus (DENV). Therefore, we assessed the vector competence of different *Aedes* strains after oral feeding on a serial titration of ZIKV (PRVABC59 strain). The mosquitoes were fed with human blood (50% v/v) and supernatants from ZIKV-infected Vero cells (50% v/v). The ZIKV load was determined from the midgut, the head, and the salivary glands over a time course. The infection, dissemination, and transmission ratios of these mosquitoes were subsequently calculated accordingly. Intriguingly, acquisition of human blood with $1 \times 10^5$ pfu/ml, but not $1 \times 10^3$–$1 \times 10^4$ pfu/ml ZIKV, resulted in a robust infection and transmission by the *A. aegypti* Rockefeller stain, the *A. albopictus* Jiangsu strain, and the *A. aegypti* Brazil strain (Supplementary Fig. 4).

**ZIKV in sewage can be acquired and transmitted by mosquitoes.** The prevalence of large ZIKV epidemics has faded. There

are a few Zika cases currently being reported in the Americas and Southern Asia[29,30]. Therefore, we propose that it is impractical to collect sewage samples from epidemic areas for the detection of infectious ZIKV. As an alternative, we collected 10 sewage samples from different cesspit locations. The characteristics of these sewage samples are shown in Supplementary Table 2. The pH values of these sewages ranged from 7.45 to 8.35; the chemical oxygen demand (COD) largely varied from 80 to 2300 mg/l; and the amount of ammonia nitrogen ($NH_3$-N) ranged from 25 to 430 mg/l (Supplementary Table 2). Subsequently, we tested the viral survivability in these sewage samples. The viral survivability in the samples was measured by a time-course plaque assay. With $1 \times 10^5$ pfu/ml of initial viral titer at 28 °C, one sample, with the highest amount of COD and $NH_3$-N (Sample #10), had completely inactivated ZIKV at the first time point of detection (1 h), while the virus in the other samples partially survived until 1 h (Sample #9) or 8 h (Samples #7 and #8) post incubation. The virus survived up to 192 h in the rest of the samples (Fig. 3a). Nonetheless, the previous literature indicates that the number of infectious ZIKV particles discharged in patient urine might be 10–20 pfu/ml[14,20]. We therefore utilized 20 pfu/ml as the initial viral titer to further assess the ZIKV stability in these sewage samples. Consistently, the samples with high concentrations of COD and $NH_3$-N (Samples #7, #8, #9, and #10) completely inactivated the viruses within 30 min

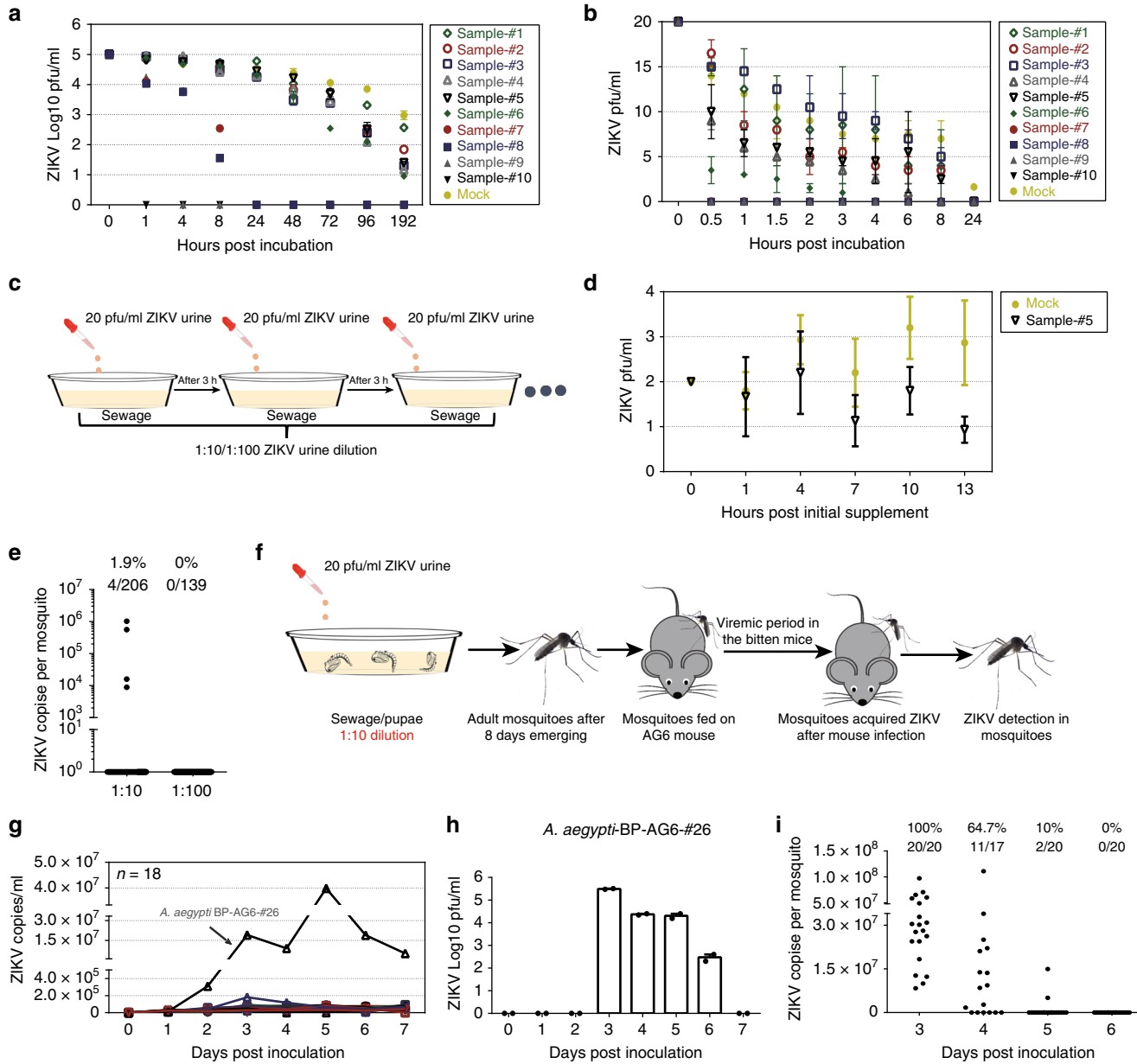

**Fig. 3** The field *A. aegypti* mosquitoes breeding in the infectious sewage transmit Zika virus (ZIKV) to hosts. **a**, **b** ZIKV survivability in the sewage samples. The supernatant from ZIKV-infected Vero cells was incubated at 28 °C with the sewage samples or phosphate-buffered saline (PBS) (Mock), with either $1 \times 10^5$ (**a**) or 20 pfu/ml (**b**) of initial titer. The viral survivability was measured over a time course by a plaque assay. **a** $n = 2$ biologically independent samples for sewages and $n = 3$ for the mock. **b** $n = 2$ biologically independent samples. **c**, **d** Continuous supplementation of ZIKV in the sewage. **c** Schematic representation. Either the sewage (Sample #5) or PBS (Mock) was used. **d** ZIKV survivability in the infectious sewage. The mixtures were maintained at 28 °C for a time course and subsequently subjected to a plaque assay. $n = 3$ biologically independent samples. **a–d** The data were repeated by two independent biological replicates with the similar results. **e** Acquisition of ZIKV infection by the *A. aegypti* Brazil Paraiba mosquitoes breeding in the infectious sewage. Both ZIKV prevalence and infectivity in *A. aegypti*. The initial ZIKV urine concentration was 20 pfu/ml. **f–i** ZIKV is transmissible between AG6 mice and mosquitoes breeding in the infectious sewage. **f** Schematic representation. **g**, **h** ZIKV viremia in the mosquito-bitten AG6 mice. **g** The presence of ZIKV genome in whole blood was assessed using quantitative PCR (qPCR) detection. $n$ represents the mouse number. **h** Detection by a plaque assay. $n = 2$ detection replicates. **i** Prevalence of ZIKV infection in the mosquitoes fed on a viremic AG6 mouse. **e**, **i** The mosquitoes for ZIKV detection by TaqMan qPCR. The number of infected mosquitoes relative to the total number of mosquitoes is shown at the top of each column. One dot represents one mosquito. The percentages are represented as the ratios of mosquito infection. **a–h** Data are presented as the mean ± SEM. **a–i** The PRVABC59 strain was used for the incubation. Source data are provided as a Source Data file

post incubation, while ZIKV maintained its survivability for more than 3 h in Samples #1 to #5 (Fig. 3b). We propose that the COD and NH₃-N concentrations in cesspit liquids may act as a key determinant for ZIKV survivability. These data indicate that ZIKV can survive in sewage under certain conditions. In the ZIKV epidemic area of America, the COD concentration of sewage varies from ~ 88 to 1323 mg/l, the pH ranges from 6.61 to 8.5, and the NH₃-N ranges from 5 to 192 mg/l, according to the field surveillance literature (Supplementary Table 3). We therefore used sewage Sample #5 (COD: 462 mg/l; NH₃-N: 85 mg/l; pH: 7.62) to mimic the sewage of the epidemic regions for mosquito breeding.

In the scenario of urination by an infected individual, the infectious urine might be discharged into a septic tank approximately every 2–3 h because a person normally urinates 6–8 times per day. To closely simulate the natural urination scenario, we continuously added either 1/10 or 1/100 dilution of 20 pfu/ml ZIKV urine to the sewage sample (Sample #5) at 3 h intervals (Fig. 3c). For continuous supplementation with a 1/10 dilution of 20 pfu/ml ZIKV urine, the viral titer was maintained at ~1–2 pfu/ml (Fig. 3d). However, the viral titer was not measurable by a plaque assay at a 1/100 dilution (the pfu was <1). We then reared the pupae of a field-derived A. aegypti Brazil Paraiba strain in these aquatic conditions. The pupae emerged in the sewage successfully (Supplementary Fig. 2b). The emerging adult mosquitoes were reared for 8 days before being subjected to ZIKV detection by RT-qPCR. Intriguingly, 1.9% (4/206) of A. aegypti adults, breeding in sewage with continuous supplementation of a 1/10 dilution of 20 pfu/ml ZIKV urine (maintenance of 1–2 pfu/ml ZIKV), showed ZIKV RNA positivity. Nonetheless, none of the mosquitoes were ZIKV-positive by quantitative PCR (qPCR) when breeding in a 1/100 ZIKV urine dilution (Fig. 3e). Consistently, 1 out of 78 mosquito pools (10 mosquitoes per pool) from the sewage with a 1/10 dilution of 20 pfu/ml ZIKV urine was positive for the isolation of infectious ZIKV in Vero cells (Table 2).

We next assessed whether the mosquitoes breeding in infectious sewage could maintain the ZIKV transmission cycle between hosts and mosquitoes. The pupae of an A. aegypti Brazil Paraiba strain were reared in the sewage maintained with 1–2 pfu/ml ZIKV. Eight days after adults emerged, 30 female A. aegypti mosquitoes were allowed to feed on an AG6 mouse. Additionally, the infected mouse was fed on by naive A. aegypti mosquitoes throughout the viremic stage. After 8 days, the fed mosquitoes were analyzed by RT-qPCR (Fig. 3f). Notably, 1 out of the 18 mice exposed to infected mosquitoes was ZIKV-positive by either RT-qPCR (Fig. 3g) or plaque assay (Fig. 3h) after mosquito feeding. The infected mouse (A. aegypti-BP-AG6-#26) died 11 days after mosquito feeding. Intriguingly, of the mosquitoes that fed on the mouse with viremia from days 3 to 5, 10–100% successfully acquired ZIKV infection (Fig. 3i). Thus, we demonstrate that the field-derived A. aegypti mosquitoes from the ZIKV epidemic areas can acquire ZIKV infection during breeding in infectious sewage in natural settings and subsequently transmit the infection to hosts by biting.

**Mouse ZIKV viruria leads to mosquito infection.** The stability of ZIKV varied dramatically in different human urine specimens (Fig. 1a, b). We next assessed key factor(s) that determine ZIKV stability in human urine. We noted that the pH values of human urine were correlated with ZIKV survivability; a neutral or slightly alkaline condition favored ZIKV survival (Table 3). Consistently, these ZIKV-hostile urine samples became favorable for ZIKV survival after their pH was adjusted to modest alkalinity with sodium hydroxide (NaOH). In contrast, decreasing the urine pH destroyed ZIKV viability (Fig. 4a). Using a larger number of urine samples, we confirmed the correlation of ZIKV stability with pH values ($r = 0.8214$, $P < 0.0001$, linear regression with correlation coefficients ($r$) and significance ($p$), Fig. 4b). Furthermore, a pH gradient assay demonstrated that a pH above 6.5 rendered ZIKV stable in human urine (Fig. 4c). Indeed, it is known that the flavivirus envelope (E) protein undergoes irreversible conformational changes at acidic pH values below 6.5, which occurs naturally during viral entry when the viral membrane fuses with the endolysosomal membrane[31,32]. This structural change in the E protein may lead to complete loss of viral infectivity by disabling viral entry. Overall, the pH value of human urine determines ZIKV infectivity and likely subsequent transmissibility to mosquitoes.

We next exploited the AG6 mouse model and A. aegypti to investigate whether ZIKV-contaminated urine could infect breeding mosquitoes. The pH in human urine varies from 4.5 to 8.0; however, mouse urine is usually more acidic, with a pH lower than 6.5[33,34]. ZIKV could not maintain its infectivity in regular mouse urine (Supplementary Fig. 5). However, after the pH value was adjusted to >6.5, the mouse urine was able to maintain ZIKV infectivity (Supplementary Fig. 5). Next, we infected the AG6 mice with ZIKV via intraperitoneal injection (100 pfu per mouse) and subsequently treated the infected mice daily with $NaHCO_3$[33,35] (Fig. 5a). The AG6 mice developed high viremia after infection (Fig. 5b). Consistent with the previous literature, $NaHCO_3$ treatment significantly elevated the pH value

---

**Table 2 Measurement of infectious ZIKV particles in mosquitoes**

| Continuous ZIKV urine supplementation | Mosquito species | Number of mosquito pools[a] | Positive pools for infectious virions |
|---|---|---|---|
| 1:10 Dilution | Aedes aegypti -Brazil Paraiba | 78 | 1 |
| 1:100 Dilution | Aedes aegypti -Brazil Paraiba | ND | ND |

ZIKV Zika virus, ND not determined
[a]Ten mosquitoes per pool

---

**Table 3 Characterization of the urine from human donors**

| Tested factors in donor urine | Donor 1 | Donor 2 | Donor 3 | Donor 4 | Donor 5 | Donor 6 | Donor 7 | Donor 8 |
|---|---|---|---|---|---|---|---|---|
| Gender | Female | Female | Female | Female | Male | Male | Male | Male |
| Color | Yellow | Light yellow | Light yellow | Yellow | Yellow | Yellow | Light yellow | Light yellow |
| Clarity | Clear | Clear | Clear | Clear | Clear | Clear | Clear | Clear |
| pH value | 6.0 | 6.6 | 7.4 | 6.4 | 5.5 | 7.2 | 6.2 | 6.8 |
| Protein (ng/µl) | 0.05 | 0.05 | 0.04 | 0.05 | 0.20 | 0.05 | 0.08 | 0.05 |
| ZIKV survivability at 37 °C (4 h pfu/0 h pfu) | 0% | 7.2% | 41% | 0% | 0% | 9.1% | 0% | 20% |
| ZIKV survivability at 28 °C (6 h pfu/0 h pfu) | 0% | 2.5% | 49.25% | 0% | 0% | 2% | 0% | 11% |

ZIKV Zika virus, pfu plaque-forming unit

---

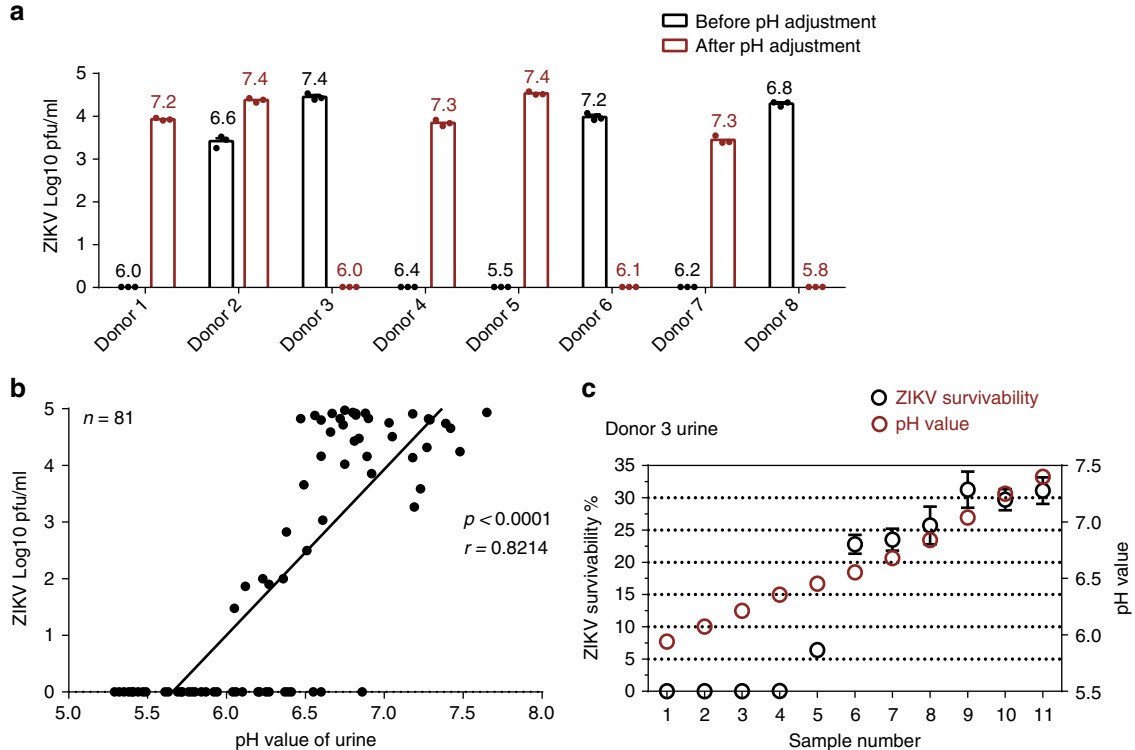

**Fig. 4** Correlation between Zika virus (ZIKV) survivability and the pH value of human urine. **a** Adjustment of pH value regulated ZIKV survivability in human urine. The pH values of human urine samples were adjusted by either NaOH or HCl. The pH value is showed at the top of each column. $n = 3$ biologically independent samples. Data are presented as the mean ± SEM. **b** ZIKV survivability in human urine correlates with the pH value of urine. One dot represents one human urine sample. Data were analyzed by linear regression with correlation coefficients (r) and significance (p). n represents the number of human urine samples. The information for the urine samples is summarized in Supplementary Table 4. **c** Determination of the correlation between ZIKV survivability and the pH value in human urine. The urine samples from Donor 3 with gradient pH values were incubated with ZIKV supernatant. The percentage of ZIKV survivability was defined as the ratio of the pfu value at 4 h incubation to the value at 0 h incubation. $n = 3$ biologically independent samples. Data are presented as the mean ± SEM. **a–c** The urine samples were incubated with ZIKV supernatant from infected Vero cells (the initial titer was $1 \times 10^5$ pfu/ml) for 4 h at 37 °C. Subsequently, the ZIKV titer was measured by a plaque assay. Source data are provided as a Source Data file

of the mouse urine (Fig. 5c). The urine pH of the NaHCO₃-treated mice started to decrease 3 days after infection, likely due to severe morbidity of the infected animals (Fig. 5c). Notably, there were almost no infectious ZIKV particles in the urine of untreated infected mice; however, 30–120 pfu/ml infectious ZIKV was observed in the urine of NaHCO₃-treated mice from days 3 to 6 post infection (Fig. 5d). The *A. aegypti* pupae were then bred in 5-fold diluted urine excreted by ZIKV-infected mice. The adult mosquitoes from pupae in the infected mouse urine were reared for an additional 8 days for RT-qPCR detection of viral loads. We found that 1.1–5.0% of the mosquitoes were positive for ZIKV RNA (Fig. 5e), demonstrating that ZIKV was transmitted from animals to mosquitoes via urine. These results indicate that ZIKV might transmit between the mosquitoes emerging from a ZIKV urine-contaminated water source and humans (Fig. 5f).

## Discussion

ZIKV is a blood-borne pathogen that is known to be transmitted by infected mosquitoes. Although mosquitoes transmit ZIKV to and acquire it from humans primarily through blood feeding, our experiments mimicking natural settings demonstrate that mosquitoes may acquire ZIKV from human urinal discharge and sewage during breeding. Our conclusions are supported by several lines of laboratory evidence. First, ZIKV may survive for up to 8 h in harsh environments, such as urinal discharge and sewage. Second, both lab-adapted and field mosquitoes were infected during hatching by ZIKV in the urinal and sewage environments at concentrations as low as 1–2 pfu/ml. Third, the adult mosquitoes that hatched from ZIKV urine/sewage were capable of transmitting ZIKV to mice and eliciting viremia. In favor of our laboratory findings, recent clinical studies have shown the presence of 10–20 infectious ZIKV particles per ml of ZIKV patient urine[14,20]. In particular, cesspits are still commonly used in areas with recent ZIKV epidemics, including Brazil and the Pacific islands[22,36,37]. Poor public sanitary infrastructure may have created a physical environment for urinal ZIKV to be acquired by mosquitoes. Nonetheless, these experimental studies attempting to mimic natural settings cannot fully represent the natural scenarios in the field. The potential limitations in experimental design, such as the number of urine and sewage samples, the experimental parameter settings (temperature, incubation time, etc.), and usage of animal models (mosquito species/strains, mouse models, etc.), suggest that the laboratory conditions may not comprehensively represent realistic conditions. Epidemiological studies will be needed to prove human viruria-mediated ZIKV infection in mosquitoes.

Our current result indicates that the maintenance of a low ZIKV titer in sewage can cause the infection of adult mosquitoes through breeding. The infected mosquitoes derived from the infectious sewage can transmit ZIKV to a host by biting. In a

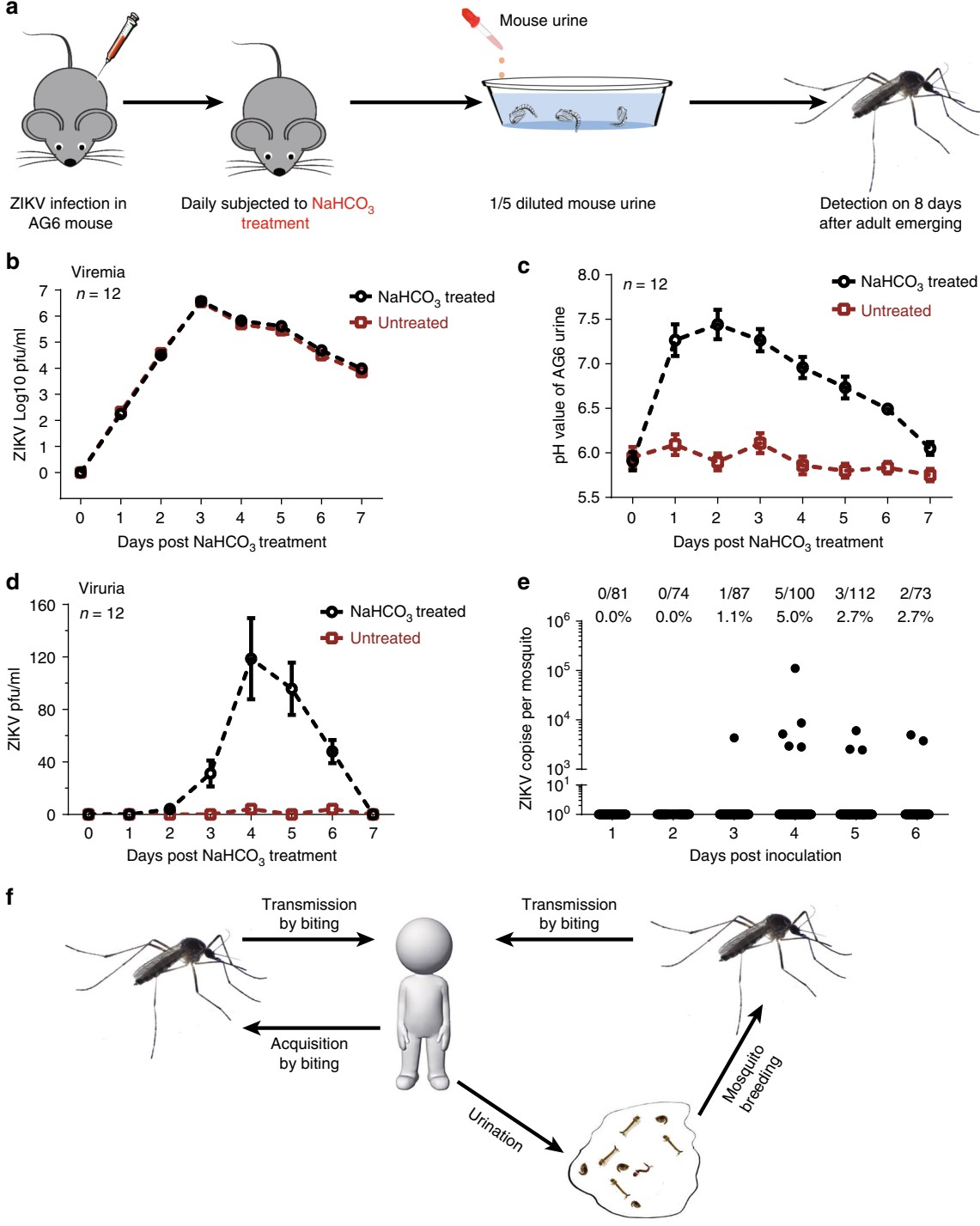

disease-outbreak scenario, many infected individuals (with or without symptoms) may intensively discharge a large volume of infectious urine into a restricted aquatic habitat of mosquitoes. We therefore speculate a scenario during a Zika outbreak in which the accumulating ZIKV infectious particles discharged by patients might be sufficient to facilitate the acquisition of an infection by immature mosquitoes breeding in cesspit conditions, thereby establishing the host urination-mosquito breeding transmission cycle. Interestingly, the emerging evidence indicates that low-passage *Aedes* field mosquitoes from the Americas have an unexpectedly low vector competence for ZIKV transmission by oral blood feeding[38]. Compared to that of CHIKV and DENV, clinical surveillance has shown a relatively low ZIKV viremia

$(10^2-10^3$ pfu/ml) present in human serum samples[8,27,28]. Intriguingly, the acquisition of human blood with $10^3-10^4$ pfu/ml ZIKV resulted in an inefficient infection and transmission by both the lab-adapted and field-derived *Aedes* mosquitoes. Thus, ZIKV viremia alone might not account for its rapid spread, implicating that additional routes might contribute to the prevalence of mosquito infection in nature. Nonetheless, human viremia titers reported in the previous literature may reflect snapshot results of patients seeking treatment after symptom onset. It is possible that these results may actually reflect off-peak viremia.

A common concept is that *Aedes* mosquitoes mainly breed in clean water. However, accumulating evidence indicates that both

**Fig. 5** Urination by infected animals leads to the infection of mosquitoes during breeding. **a** Schematic representation of mosquito infection by breeding in diluted infected mouse urine. We infected the AG6 mice by Zika virus (ZIKV) intrapleural injection (100 pfu per mouse) and subsequently subjected the infected mice to daily NaHCO$_3$ treatment. The infected AG6 mice without NaHCO$_3$ treatment served as untreated controls. The urine was collected and subsequently diluted 5-fold for the *A. aegypti* (Rockefeller strain) pupal eclosion. The emerging mosquitoes were reared for an additional 8 days for quantitative reverse transcription PCR (RT-qPCR) detection. **b** ZIKV viremia in the infected AG6 mice. The viral load was detected by a plaque assay over a time course after infection. **c** Regulation of the pH value in the urine of NaHCO$_3$-treated infected AG6 mice. **d** Assessing the ZIKV infectious particles in the urine of infected AG6 mice. The urine samples collected daily were measured by a plaque assay. **b–d** *n* represents 12 independent mice in each group. The data were pooled from three independent experiments. Data are presented as the mean ± SEM. **e** Mosquitoes acquired ZIKV infection by breeding in diluted urine from NaHCO$_3$-treated infected AG6 mice. The urine samples from four individual mice were mixed and then diluted 5-fold for *A. aegypti* pupal eclosion. The emerging adults were reared for 8 days for ZIKV detection by RT-qPCR. The number of infected mosquitoes relative to the total number of mosquitoes is shown at the top of each column. One dot represents one mosquito. The percentages are represented as the ratios of mosquito infection. The data were pooled from three independent biological replicates. **f** Schematic representation of the host urination-mosquito breeding ZIKV transmission route. A Zika patient may release urine daily, resulting in various levels of infectious ZIKV being excreted by patient urination in natural circumstances. The mosquitos breeding from the water systems contaminated by ZIKV may acquire the infection. Therefore, viruria of infected hosts may facilitate ZIKV prevalence and transmission by mosquitoes. **b–e** Source data are provided as a Source Data file

*A. aegypti* and *A. albopictus* have evolved to oviposit and breed in wastewater with low oxygen and high turbidity. Notably, the population of adult *A. aegypti* mosquitoes emerging from sewage may be equal to or even larger than those emerging from traditional breeding sites (clean water) in Brazil[23] and Puerto Rico[24]. Nonetheless, the epidemiological importance of cesspit breeding for ZIKV transmission remains unclear. Further field surveillance is therefore needed to investigate the breeding behavior of these *Aedes* spp. in cesspit conditions. In this study, we selected 28 °C as a temperature parameter in our laboratory setting to mimic the natural conditions. Indeed, the mosquitoes maintained in the laboratory were routinely reared at 28 °C as a standard condition[25,39,40]. The ZIKV epidemic area in Brazil located in tropical regions, where the maximum daytime temperature is 28.6–32.8 °C and the minimum temperature ranges from 20.5–22.3 °C at night[41]. We noted that the temperature setting at 28 °C might represent a realistic daytime condition in the ZIKV epidemic area. Nonetheless, nighttime temperatures are also essential for mosquito breeding and are likely to be substantially lower. Therefore, both mosquito breeding and ZIKV stability under various temperature conditions remain to be further investigated.

ZIKV is well known to maintain a transmission cycle between its hosts and mosquitoes primarily through mosquito blood-feeding behavior. We report here that mosquitoes breeding from a water system contaminated by ZIKV-containing human urine can carry and transmit ZIKV. Our study suggests that mosquitoes might acquire ZIKV not only through feeding on a viremic host[25] but also through urine excreted by ZIKV patients. In addition to ZIKV, the RNA of other flaviviruses, such as DENV[26,42], yellow fever virus[43], and West Nile viruses (WNV)[44] have also been found in infected human urine. The presence of infectious particles has been detected in the urine of WNV patients[45], suggesting that other flaviviruses in human urine might also be transmissible. The mosquito-borne flaviviral diseases widely occur in the tropical and subtropical regions of developing countries with a rudimentary sewer system and poor basic sanitation conditions[46,47]. ZIKV-containing human urine may contaminate the water systems where *Aedes* mosquitoes breed[48]. This study offers laboratory evidence that ZIKV could be transmitted to mosquitoes via human urine and provides a possible approach for preventing the dissemination of mosquito-borne flaviviral diseases by interrupting the ZIKV transmission cycle.

## Methods
**Ethics statement**. Human urine was collected from healthy donors who provided informed written consent. The collection of human urine samples was approved by the local ethics committee of Tsinghua University.

**Mice, mosquitoes, cells, and viruses**. C57BL/6 mice deficient in type I and II IFN receptors (AG6 mouse) were donated by the Institute Pasteur of Shanghai, Chinese Academy of Sciences. The mice were bred and maintained in a specific pathogen-free animal facility at Tsinghua University. All animal protocols used in this study were approved by the IACUC (Institutional Animal Care and Use Committee) of Tsinghua University and performed in accordance with the IACUC guidelines. The laboratory animal facility is accredited by the AAALAC (Association for Assessment and Accreditation of Laboratory Animal Care International). Groups of age- and sex-matched AG6 mice, 6–8 weeks of age, were used for the animal study. The *A. aegypti* Rockefeller strain, the *A. albopictus* Jiangsu strain, the *A. aegypti* Brazil Paraiba strain, and the *C. quinquefasciatus* Hainan strain were reared in a low-temperature, illuminated incubator (Model 818, Thermo Electron Corporation, Waltham, MA, USA) at 28 °C and 80% humidity according to standard rearing procedures[25,49,50]. The DENV2 New Guinea C strain (*M29095*), ZIKV PRVABC59 strain (*KU501215*), BATV YN92–4 strain[51], and SINV MRE-16 strain (*U90536.1*) were grown in Vero cells for viral production[25,49,52]. The Vero cells were maintained in Dulbecco's modified Eagle's medium (DMEM, C11965500BT, Gibco), supplemented with 10% heat-inactivated fetal bovine serum (16000-044, Gibco). The Vero cell line was purchased from ATCC (CCL81). The cell lines had no mycoplasma contamination. All of the viruses were titrated by a plaque assay[53]. Briefly, the titrated samples were added to 95–100% confluent Vero cells in 6-well plates, and then incubated at 37 °C and 5% CO$_2$ for 4 h. The wells were washed with PBS, and subsequently overlaid with 2.5 ml of DMEM-agarose. After 5–6 days, the cells were stained with 1% crystal violet. The plaques were counted for viral particle calculation.

**Urinalysis**. The characterization of human urine samples, such as the specific gravity, clarity, and color, was measured by a Fully Automated Urinalysis System (AX-4030, ARKRAY). The pH value of human urine was measured using a basic pH meter (PB-10, Sartorius). The urinalysis results are summarized in Supplementary Table 4. The mouse urine was discharged by abdominal pressing, collected by micropipette, and transferred to a 1.5 ml tube for further investigation. The pH value of mouse urine was measured by pH testing strips (90304, Macherey Nagel).

**Sewage analysis**. Both COD and NH$_3$-N were measured in the sewage samples. The COD value was detected by COD digestion vials (2125915-CN, HACH) and a portable spectrophotometer (DR1900, HACH). The NH$_3$-N was measured with a Nitrogen-Ammonia reagent set (2606945-CN, HACH) and a portable spectrophotometer (DR1900, HACH). The pH value of sewage samples was measured using a basic pH meter (PB-10, Sartorius). The sewage characteristics are summarized in Supplementary Table 2.

**Mosquito infection by breeding in ZIKV-containing urine**. Fresh human urine from Donor 3 was mixed with ZIKV supernatant from infected Vero cells. The mixture is named ZIKV urine hereafter. ZIKV urine was used for breeding the pupae of *A. aegypti* (Rockefeller strain, Brazil Paraiba strain) and *A. albopictus* (Jiangsu strain). After exposure to ZIKV urine, the larvae or pupae were transferred to a container with fresh water until eclosion. The emerging mosquitoes were reared for an additional 8 days for viral detection.

**Viral genome quantitation by TaqMan qPCR**. Total RNA was isolated from homogenized mosquitoes using an RNeasy Mini Kit (74106, Qiagen) and reverse transcribed into complementary DNA (cDNA) using an iScript cDNA Synthesis Kit (170-8890, Bio-Rad). Viral genomes were quantified via qPCR amplification of ZIKV, DENV, BATV, and SINV genes. The viral genomic copies were normalized by a standard curve[40]. In each subsequent RT-qPCR plate sample, we quantified

four standard aliquot dilutions to adjust for threshold variation between plates. The primers and probes used for this analysis are shown in Supplementary Table 5.

**ZIKV detection and viral isolation from mosquito pools**. The freshly emerging pupae of *A. aegypti* and *A. albopictus* were reared in ZIKV urine. The freshly emerging pupae of *A. aegypti* were reared in either human urine or sewage with ZIKV urine. Eight days after emerging, ten adult mosquitoes were pooled and ground in 400 μl PBS buffer (C10010500, Gibco). The homogenized mosquitoes were then centrifuged at $20,000 \times g$ for 5 min at 4 °C. The supernatant was then filtered by a 0.22 μm filtration unit (SLGV013SL, Millipore) for culturing in Vero cells. Total RNA was isolated from the infectious supernatant of Vero cells using an RNeasy Mini Kit (74106, Qiagen) and reverse transcribed into cDNA using an iScript cDNA Synthesis Kit (170-8890, Bio-Rad). We amplified ZIKV fragments by PCR. The PCR products were subjected to viral sequencing. The sequencing primers are shown in Supplementary Table 5.

**Viral survivability in sewage samples**. Human urine (Donor #3) with 20 pfu/ml ZIKV (PRVABC59 strain) was added to the sewage (Sample #5) or PBS, as a mock control, at either 1:10 or 1:100 dilutions. Urine with 20 pfu/ml ZIKV was continuously supplemented into sewage or PBS at 3-h intervals. The infectious sewage or PBS was incubated at 28 °C. Viral survivability was determined at 1 h post ZIKV supplementation via a plaque assay. Briefly, the infectious sewage was filtered by a 0.22 μm filtration unit (SLGV013SL, Millipore) to remove bacterial contamination. We next used 5 ml of filtered infectious sewage to add into a well of 6-well plate for a plaque assay. After 4 h of incubation with Vero cells, the infectious sewage samples were removed and replaced with 2% DMEM (containing 1% agarose) for 5 days. The sample was measured twice.

**Mosquito feeding on AG6 mice**. The pupae of *A. aegypti* (Rockefeller strain) and *A. aegypti* (Brazil Paraiba strain) bred in ZIKV urine. Eight days after adult mosquitoes emerged, 20 emerged female mosquitoes were separated into a netting-covered container for blood feeding. The mosquitoes were starved for 24 h before blood engorgement. Six- to eight-week-old AG6 mice were anesthetized and placed on the top of the containers. The mosquitoes were allowed to feed on the mice for 30 min in darkness. The blood samples were collected daily from the tail veins of AG6 mice, and the viral genome was quantified via TaqMan qPCR amplification of the ZIKV E gene. The number of infectious ZIKV particles in plasma at the viremic period was determined by a plaque assay.

**ZIKV transmission by mosquitoes breeding in sewage**. Pupae of *A. aegypti* (Brazil Paraiba strain) bred in the sewage with continuous supplementation of ZIKV urine. Eight days after adult mosquitoes emerged, 30 emerged female mosquitoes were separated into a netting-covered container for blood feeding. The mosquitoes were starved for 24 h before blood engorgement. The 6–8-week-old AG6 mice were anesthetized and placed on the top of the containers. The mosquitoes were allowed to feed on the mice for 30 min in darkness. Blood samples were collected daily from the tail veins of AG6 mice, and the viral genome was quantified via TaqMan qPCR amplification of the ZIKV E gene. The infectious ZIKV particles in plasma during the viremic period were determined by a plaque assay. Subsequently, the infected mice were fed on by naive *A. aegypti* on the viremic day. The fed mosquitoes were reared for an additional 8 days for RT-qPCR viral detection.

**Infected AG6 mouse viruria model**. AG6 mice (6–8 weeks) were infected by ZIKV intrapleural injection (100 pfu per mouse). The infected AG6 mice were administered 200 mM NaHCO$_3$ solution in regular drinking water[33,34]. One hour before urine collection, the AG6 mice were intragastrically administered 500 μl of 200 mM NaHCO$_3$. Mouse urine was collected by abdominal pressing.

**Determination of viremia in infected mice by a plaque assay**. For viremia detection, blood samples from the tail veins of infected mice were collected in 0.4% sodium citrate and centrifuged for 5 min at $6000 \times g$ and 4 °C for plasma isolation. The presence of infectious viral particles in plasma was determined via a plaque assay[53]. Briefly, the titrated plasma samples were added to 95–100% confluent Vero cells in 6-well plates, and then incubated at 37 °C and 5% CO$_2$ for 4 h. The wells were washed with PBS, and subsequently overlaid with 2.5 ml of DMEM-agarose. After 5–6 days, the cells were stained with 1% crystal violet. The plaques were counted for viral particle calculation.

**Statistics**. Animals were randomly allocated to different groups. Mosquitoes that died before measurement were excluded from the analysis. The investigators were not blinded to the allocation during the experiments or to the outcome assessment. No statistical methods were used to predetermine the sample size. Descriptive statistics are provided in the figure legends. A Kruskal–Wallis analysis of variance was conducted to detect any significant variation among replicates. If no significant variation was detected, the results were pooled for further comparison. Linear regression analysis was used to assess the correlation between ZIKV survivability

and the urine pH value in human urine. All analyses were performed using the GraphPad Prism statistical software.

**Reporting summary**. Further information on experimental design is available in the Nature Research Reporting Summary linked to this article.

## Data availability

All data generated or analyzed during this study are included in this published article and its Supplementary information files. The source data underlying Figs. 1a–e, 2a–f, 3a–e, 3g–i, 4a–c, 5b–e, and Supplementary Figures 1b, 2a–b, 3a–f, 4a–c and 5 are provided as a Source Data file.

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

## Acknowledgements

This work was funded by grants from the National Key Research and Development Plan of China (2018ZX09711003-004-003, 2018YFA0507202, 2016YFD0500400, 2017ZX10305501-003), the Natural Science Foundation of China (31825001, 81730063, 81571975, 81673234), the Shenzhen San-Ming Project for prevention and research on vector-borne diseases (SZSM201611064), and the Municipal Healthcare Joint-Innovation Major Project of Guangzhou (201704020229). P.W. is supported by a National Institutes of Health award R01AI132526. We thank that Professor Scott Weaver from the UTMB provided critical suggestions for the manuscript. G.C. is a Newton Advanced Fellow awarded by the Academy of Medical Sciences and the Newton Fund. We also thank the core facilities of the Center for Life Sciences and Center of Biomedical Analysis for technical assistance (Tsinghua University).

## Author contributions

G.C. designed the experiments. G.C. wrote the manuscript; S.D., Y.L., and J.L. performed the majority of the experiments and analyzed the data; J.Z. and L.T. assisted in RNA isolation and qPCR detection. G.D.L. provided the viral strains. P.M. provided the human urine samples. C.-H.C. and Q.Y.L. contributed in the mosquito rearing. R.Z., C.C., B.C., F.C.Z., C.-F.Q., P.-Y.S., P.W., and H.T. contributed to experimental suggestions and strengthened the writing of the manuscript. All authors reviewed, critiqued, and provided comments on the text.

## Additional information

**Competing interests:** The authors declare no competing interests.

