## [Peer Review File · Nature Communications]

Reviewers' Comments:

Reviewer #1:

Remarks to the Author:

In the manuscript 'Viruria facilitates Zika virus prevalence and transmission by mosquitoes' by Liu et al., the authors investigate a novel model/concept of how Zika virus (ZIKV) present in human urine may contribute to infection of naïve *Aedes aegypti* mosquitoes. The authors conclude that ZIKV may be shed into sewage waters with the urine of infected patients, and that ZIKV can infect developing larvae/pupae of *Aedes aegypti* mosquitoes and result in the emergence of mosquitoes that are able to transmit ZIKV without first obtaining an infectious bloodmeal. While the study concept is novel and intriguing, the detailed rationale, design, methodology and conclusions of the study are unfortunately flawed in a number of aspects. The main concern is with the authors' broad conclusions that are not supported by the study.

The concept that mosquito larvae can acquire virus from the water is interesting, and if this was presented as the major conclusion of the paper without the other grand conclusions about viruria contributing to ZIKV transmission, the manuscript would be significantly more suitable for publication. The experimental results of the study are intriguing but do not provide enough evidence to support the overall conclusions of the paper.

My initial major concern with the study is related to its relevance – the overall idea is interesting, but also somewhat far-fetched and the rationale provided in the manuscript background seems flawed. While the authors provide a surprising amount of evidence (citations) that mosquitoes have been shown to breed in environments which contain urine, such as septic tanks and cesspits, these are most likely entirely different environments compared to urine diluted in water in the lab, as used in this study to 'mimic' cesspit/sewage conditions. The presence of microorganisms, macroorganisms, human fecal matter and (depending on the specific environment) exposure to UV will probably severely impact virus stability beyond just urine, water and temperature. In fact, reference 23 highlights that the low-turbidity water sampled from cesspits was around a pH of 5.3, which the authors have shown themselves results in rapid loss of ZIKV infectivity. pH conditions will most likely vary significantly between these harsh environments and will influence stability. In addition, chemicals present in these rudimentary sewage systems may influence virus stability. One way to improve this aspect of the study would be to either test sewage samples in endemic/epidemic areas for the presence of infectious ZIKV (not just RNA), or to perform a controlled stability experiment in the lab using sewage samples to incubate ZIKV (ideally from various sources and at different temperatures/pH levels). If these non-sterile, harsh environments will not result in a significant loss of infectivity, mosquito rearing in the presence of ZIKV should be attempted. These studies could improve the potential relevance and credibility of this study as it relates to the field.

However, an additional major concern is the assumption that significant amounts of infectious virus are shed with urine of infected patients. The authors cite references to support the notion that urine of ZIKV infected patients contains high viral loads – however, these viral loads are based on RNA, not infectious virus. The statement in line 79-80 in which the authors say '...discharge high levels of infectious ZIKV (10^5 - 10^6 copies/mL)' is highly misleading – the copy numbers are correct and have been reported, but again these are NOT necessarily infectious, and instead these 'viral loads' are based purely on viral RNA. The only study in which virus was titrated directly from a urine samples by plaque assay (reference 14) found only a single plaque in one sample and suggested an infectious viral titer of only 10 PFU/mL in the urine of this patient. All other tested urine samples were negative (or plaque titers were not determined). While other studies which isolated ZIKV from urine, may have involved samples with higher viral titers, these data were not provided in any of the cited publications. Thus, there is no convincing evidence for urine samples containing 10^3 PFU/mL or even 20 PFU/mL

as used in some experiments. In cell culture, a few infectious virus particles may be sufficient to infect the culture and result in virus isolation. Without the acquisition of human urine samples of infected patients and titration of ZIKV from urine samples, this question will remain a major concern for the relevance of the presented study.

Finally, the modelling aspect of this study seems artificial and the estimated numbers of 12-16% of naïve mosquitoes acquiring ZIKV through contaminated urine highly unlikely based on the data presented (in which only few mosquitoes acquired ZIKV during development, see Fig1h). Without clear information on mosquito breeding in cesspits/septic tank in these locations, as well as actual data on infectious virus levels in the urine of ZIKV infected patients over time, these models cannot be relied upon. In addition, not enough evidence was given on the actual parameters used in the study (what numbers were used for e.g. extrinsic and intrinsic incubation periods), and the fact that sexual transmission was not included in the model will most likely confound results as well.

Minor concerns:

Another concern that is somewhere between major and minor is the evaluation of ZIKV positive/negative mosquitoes (Fig 1e/f). The presentation of ZIKV RNA relative to actin mRNA is not very useful for interpretation by the reader. Viral copy numbers estimated from a ZIKV standard curve would allow for much better evaluation of the data. Since ZIKV copy numbers were shown later on in Fig 2b/e, it is confusing why another way of presenting ZIKV RNA levels was shown here. Furthermore, the numbers of mosquitoes that were in fact positive for infectious virus after exposure to ZIKV during development were very low (2/68 *Ae. aegypti* pools and 1/54 *Ae. albopictus* pools), suggesting that infection through larval water was in fact very inefficient. The minimum infection rate for *Ae. aegypti* was only 0.3% and the maximum infection rate 2.9%. However, this aspect is presented in a very misleading way throughout the manuscript. On a positive note, a low PFU/mL titer of 20 PFU/mL was used in these experiments, which may be slightly more relevant to field conditions than 1000 PFU/mL (although still, we have no evidence for any specific titer in sewage water).

Some small grammar/English errors throughout the manuscript should be addressed (e.g. line 88-80: the use of 'Although' and 'however' is redundant; line 90: unsure what is meant by 'have habited')

Figure 1a: The description makes this sound as if it's the experimental design, while really this is what is hypothesized to occur in nature, and what was mimicked in the lab. This is partially due to the use of past tense – the wording should be adjusted to improve clarity of the figure legend.

Figure 1b/c: In these experiments, natural conditions of the bladder and natural habitats are being mimicked. Using 10^5 PFU/mL for the experiment in Fig 1b seems somewhat reasonable, since this much virus may be secreted into the bladder. However, the authors also used 10^5 PFU/mL for urine diluted 1:50 in water (Fig.1c) and there is no evidence for any ZIKV patients shedding anywhere near 10^5 PFU/mL ZIKV in urine. The starting titer for this experiment should have been significantly lower.

Supplementary 1: Some of the text for reference 26 and 30 is cut off (size of table row too small?).

Reviewer #2:

Remarks to the Author:

In the proposed paper, the authors explore the extent to which ZIKV in urine can be transmitted to mosquitoes. They then incorporate urine driven mosquito ZIKV infection into mathematical models to

help explain the epidemics in French Polynesia and Yap Island. That infectious ZIKV particles can exist in urine is known (Zhang et al., Bonaldo et al.,). That mosquitoes can get infected from urine is novel and interesting, however, it was unclear to me whether the concentrations used in the study were epidemiologically relevant. I also found the evidence for urine-mediated transmission being epidemiologically relevant to be weak. The authors state that 15% of mosquitoes in three outbreaks acquired infection from urine, however, this was made using simplistic models with no consideration of the epidemiological situation. Detailed comments below.

The authors use 10^3 PFU/ml for the initial concentration in urine, which seems high. Bonaldo et al (PLoS NTD 2016) found 10 pfu/ml. Lustig et al., had similar number (10-20 pfu/ml). Gourinat et al., tried but failed to isolate infectious particles from urine, suggesting that RNA but not whole virus was present. The 50-fold dilution used by the authors will still leave similar or even higher levels of virus than these other studies. The rationale for the 1:50 dilution is unclear. The authors may want to consider the effect of more diluted urine samples. Given that the proportion of infected mosquitoes was already very low ($\sim 1\%$ in 1:50 dilution versus 4% in undiluted), there may no longer be transmission. This is an important consideration as if urine to mosquito transmission requires high concentrations, this is unlikely to be epidemiologically relevant.

The mosquitoes in the island outbreaks considered in the models were not the ones considered in their experiments. In French Polynesia, *Aedes polynesiensis* mosquitoes appeared to have an important role. In Yap, *Aedes Hensili* seemed key with very little *Aegypti* present (see Ledermann, PLoS NTD, 2014). The relevance of the experimental data in describing these outbreaks is therefore unclear. It is also unclear which species the fixed parameters in the model are fit to (including mosquito lifespan, extrinsic incubation period etc).

A further key issue with the model is that there is no assessment of whether the models that include urine-mediated transmission outperform models that do not include it. ϕ , the water contamination rate is very close to 0 (0.0003) indicating that W is very small which suggests there maybe little differences between models that do and do not include a urine component, however, this should be assessed.

There is no information on the priors used – especially on the key ϵ and ϕ parameters. For some of the parameters in the original model that the authors rely on, the posterior is highly dependent on the prior (see Figures 10-13 in Champagne et al., eLife) – it wouldn't be surprising if the priors had an important role with these two additional parameters too.

Minor points:

It is unclear which model the authors use – the original papers presented a number of competing models.

The model uses a vertical transmission rate of 0.25 based on a reference. However in the reference, the authors estimate that 1:290 vertical transmission proportion for *Aegypti* and 0 for *albopictus*. The authors should clarify how they obtained their estimate. Again, values for *polynesiensis/hensili* are not available.

In a number of places in the introduction, the authors state that ZIKV develops a high viremia. However, compared to CHIKV and DENV, viremia in ZIKV patients seems generally low (1-2 logs lower) (see e.g., Waggoner et al., CID 2016).

The authors state that "consistently, the two field mosquitoes all acquired ZIKV infection during

incubation" (line 155) – this is misleading as written as only a very small proportion become infected.

Reviewer #3:

Remarks to the Author:

Summary: this manuscript by Liu et al. provides experimental evidence that Zika virus-positive urine could serve as a source of virus for acquisition by mosquitoes. In this study, the authors determined the stability of Zika virus in different urine samples and under different environmental (but laboratory) conditions. The authors then reared *Aedes aegypti*, *Ae. albopictus*, and *Culex quinquefasciatus* in water spiked with ZIKV-infected urine and assayed for infection via QRT-PCR and cell culture. In addition, mosquitoes that were reared in urine water were allowed to feed on Type I and II, IFN-deficient mice and transmission of Zika virus was demonstrated, implicating ZIKV-positive urine as a previously unidentified source for mosquito infection. While these results are interesting and worthy of follow up, enthusiasm for the conclusions drawn from this work are diminished by 1.) the experimental set up, which is nowhere close to biologically relevant both in terms of the volume of water that mosquitoes were reared in as well as the complete absence of other organic material that might be present in waste water that might also inactivate Zika virus (or influence pH), and 2.) appropriate controls demonstrating the peroral vector competence of their mosquito strains for ZIKV are missing, i.e., it is important to know what the infection, dissemination, and transmission rates of these mosquitoes are when exposed to a ZIKV-positive bloodmeal. Furthermore, the *Aedes aegypti* strains tested were the Rockefeller strain, which is highly lab adapted and two Asian strains. Critically, there were no American *Aedes aegypti* included in the study. Finally, the model does not provide convincing evidence that ZIKV-positive urine is a source of ZIKV infection for mosquitoes.

Grammar and syntax need to be improved throughout.

Line 44 and 64, stating that ZIKV is maintained between mosquitoes and primates is misleading. It still is not clear how ZIKV is maintained in sylvatic transmission cycles, evidence suggests nonhuman primate involvement but it is not definitive.

Line 45, clinical studies have shown that ZIKV RNA persists in urine for an extended period of time. It is less clear how long infectious virus is shed in urine. The use of 'living infectious virus' is inaccurate, please revise.

Line 66, please include references to confirm that ZIKV develops high viremia in human blood. Most studies measure vRNA loads in human samples.

Line 76 replace "living" with "infectious".

Line 79, the authors claim that high levels of infectious ZIKV are discharged in the urine but then list a range of vRNA copies not PFU. How much infectious ZIKV is discharged with urine, it is likely at least 1000-fold lower than the amount of vRNA detected. Again, it is disingenuous to state that viremia persists for a long time. vRNA can persist for extended periods in urine but how long is it infectious? Also, for ZIKV shed in semen it has been shown that the PFU:particle rate fluctuates drastically over time. Does this also happen with urine?

Line 78, If a single human is shedding ZIKV in urine into gallons of waste water will the virus be diluted out to the point it won't matter? Given the variability reported later in the study about the heterogeneity of ZIKV stability in urine samples from different individuals, might ZIKV not be inactivated once introduced into the milieu of urine samples present in waste water?

Line 87-96 The scientific premise is faulty. Mosquito productivity depends on myriad factors, including the nutritional quality of the larval environment, container type, surrounding environmental conditions, etc. While it is generally accepted that *Ae. aegypti* prefer to oviposit in clean water, there have been documented cases of immature development in polluted water and raw sewage as demonstrated by Supplementary Data 1. But it is not clear how epidemiologically important these types of containers are for oviposition for *Ae. aegypti* or *Ae. albopictus*. It would be a potentially very different situation if the primary vector for ZIKV were, for example, *Armigeres subalbatus*, which is known to breed in sewage. However, this mosquito species is not found in the New World.

Line 113-127, what volume of water were the aquatic stages reared in and what was the overall infectious ZIKV titer in this volume of water and how does this relate to what one might expect to find in a septic tank, for example? Really, what I am asking is how biologically relevant is this in terms of a route of transmission to mosquito larvae or pupae? Did you try to estimate the minimum infectious dose for the larvae and pupae? Is this really the best experiment? It certainly would have been more stringent to include additional ZIKV-negative urine samples mixed in as a follow up as well as other biologic material that might impact ZIKV infectivity.

Line 125, it is surprising that the pupal stage was the most susceptible. What is the proposed explanation since pupae do not eat and breathe through respiratory trumpets?

Line 129-149 and 168-188, How were the adult mosquitoes processed or handled to eliminate the possibility of surface contamination? While I acknowledge that surface contamination is unlikely given the time frame of testing adults after emergence, testing for infectious ZIKV in pools of adults indicate a very low infection rate (~0.2%) suggesting that QRT-PCR is detecting residual vRNA that is not actually infectious. And how do you reconcile this very low infection rate in your pools with a fairly moderate transmission rate to mice? Again, I question the biological relevance. The mouse model used is very similar to the AG129 mouse model in which it has been shown that less than 1 PFU can cause high viremia and lethal disease in that model (see Salvo et al. 2018 PLoS NTD). It remains unclear how much ZIKV needs to be delivered by a feeding/probing mosquito to initiate infection in a human. Finally, all of the mouse serum was screened via QRT-PCR. It would be informative to verify that there is actually replicating virus in these animals.

Line 161, were the 2 DENV-2 positive mosquitoes positive by QRT-PCR or via some other method to detect infectious virus?

Line 192, correlated should be positively (or negatively) correlated measured by a statistical assay.

Line 228, it is unclear whether the model is parameterized correctly. The most plausible vector for ZIKV during the Yap outbreak was *Aedes hensilli* not *Aedes aegypti*, so the findings of this manuscript are not directly relevant. I don't believe *Aedes aegypti* have been found on Yap. There are two potential vectors in French Polynesia: *Aedes aegypti* and *Aedes polynesiensis*. During the 2013 outbreak over 2000 female *Ae. Aegypti* were collected and tested for ZIKV and only a single pool was positive. Likewise, what are the sanitation systems like on these islands and are they likely to produce large numbers of mosquitoes?

Line 244, the use of 'life cycle' is incorrect, it should be transmission cycle.

Dear Editors,

Thank you for the comments on our manuscript (NCOMMS-18-19357) titled "Viruria facilitates Zika virus prevalence and transmission by mosquitoes". We have taken all of the comments into consideration and have revised our manuscript accordingly. We believe that our additional experiments, revised analyses, and modifications of the text have substantially improved the manuscript.

We were encouraged by the reviewers' positive comments, such as "While the study concept is novel and intriguing...." (Reviewer #1), "That infectious ZIKV particles can exist in urine is known (Zhang et al., Bonaldo et al.)", "That mosquitoes can get infected from urine is novel and interesting....." (Reviewer #2) and "These results are interesting and worthy of follow up....." (Reviewer #3). The reviewers also provided important suggestions to improve the manuscript. We have included new experimental data in our manuscript per the reviewers' suggestions. Along with this letter, we have provided point-by-point responses to the questions and suggestions proposed by the reviewers.

Responses to Referee #1:

#1A. In the manuscript 'Viruria facilitates Zika virus prevalence and transmission by mosquitoes' by Liu et al., the authors investigate a novel model/concept of how Zika virus (ZIKV) present in human urine may contribute to infection of naïve *Aedes aegypti* mosquitoes. The authors conclude that ZIKV may be shed into sewage waters with the urine of infected patients, and that ZIKV can infect developing larvae/pupae of *Aedes aegypti* mosquitoes and result in the emergence of mosquitoes that are able to transmit ZIKV without first obtaining an infectious bloodmeal. While the study concept is novel and intriguing, the detailed rationale, design, methodology and conclusions of the study are unfortunately flawed in a number of aspects. The main concern is with the authors' broad conclusions that are not supported by the study.

The concept that mosquito larvae can acquire virus from the water is interesting, and if this was presented as the major conclusion of the paper without the other grand conclusions about viruria contributing to ZIKV transmission, the manuscript would be significantly more suitable for publication. The experimental results of the study are intriguing but do not provide enough evidence to support the overall conclusions of the paper.

#1A Answer: Thanks for the reviewer's general comments. We recognize that although our laboratory conditions do not completely represent the complex nature of human-virus-environment interactions, at a minimum provide proof-of-principle for ZIKV transmission via the urine-mosquito route. We have included new experimental data in our manuscript per the reviewers' suggestions.

The major changes are:

1. We demonstrated that different *Aedes* mosquito species are susceptible to ZIKV infection from breeding in human urine with lower ZIKV titers (20 pfu/ml and 2 pfu/ml) (Fig. 1d and e).

2. We found that either the *A. aegypti* Rockefeller strain or the field *A. aegypti* mosquitoes (Brazil Paraiba strain) breeding in human urine with a lower ZIKV titer (20pfu/ml) can transmit ZIKV to a naïve host (Fig. 2b-c and e-f).

3. We assessed the vector competence of different *Aedes* strains after feeding on a serial titration of ZIKV (Suppl. Fig. 4).

4. We demonstrated that the field-derived *A. aegypti* mosquitoes (Brazil Paraiba strain) from the ZIKV epidemic areas can acquire ZIKV infection during breeding in infectious sewage in natural settings and subsequently transmit the infection to hosts by biting (Fig. 3).

5. The parameters for mathematic modeling was optimized based on the data from the Brazil Paraiba strain to reflect the epidemiological relevance (Fig. 6a-d and suppl. Table 5).

Based on the reviewer's suggestions, we re-designed the experiments to closely simulate the natural settings. We hope that our additional experiments may substantially improve the revised manuscript. We have provided point-by-point responses to the questions and suggestions proposed by the reviewers.

#1B. *My initial major concern with the study is related to its relevance – the overall idea is interesting, but also somewhat far-fetched and the rationale provided in the manuscript background seems flawed. While the authors provide a surprising amount of evidence (citations) that mosquitoes have been shown to breed in environments which contain urine, such as septic tanks and cesspits, these are most likely entirely different environments compared to urine diluted in water in the lab, as used in this study to 'mimic' cesspit/sewage conditions. The presence of microorganisms, macroorganisms, human fecal matter and (depending on the specific environment) exposure to UV will probably severely impact virus stability beyond just urine, water and temperature. In fact, reference 23 highlights that the low-turbidity water sampled from cesspits was around a pH of 5.3, which the authors have shown themselves results in rapid loss of ZIKV infectivity. pH conditions will most likely vary significantly between these harsh environments and will influence stability. In addition, chemicals present in these rudimentary sewage systems may influence virus stability. One way to improve this aspect of the study would be to either test sewage samples in endemic/epidemic areas for the presence of infectious ZIKV (not just RNA), or to perform a controlled stability experiment in the lab using sewage samples to incubate ZIKV (ideally from various sources and at different temperatures/pH levels). If these non-sterile, harsh environments will not result in a significant loss of infectivity, mosquito rearing in the presence of ZIKV should be attempted. These studies could improve the potential relevance and credibility of this study as it relates to the field.*

#1B Answer: We fully agree with the reviewer's concern. The experimental

conditions in the original studies (breeding mosquitoes in urine-diluted water) cannot represent cesspit/sewage conditions in nature. The reviewer has suggested two alternative experiments to address this concern, either testing infectious ZIKV sewage samples from the ZIKV epidemic areas or assessing viral survivability in sewage samples incubated with ZIKV. The prevalence of large ZIKV epidemics has faded away. There are a few Zika cases reported in the Americas and Southern Asia (CDC, Available at: <https://www.cdc.gov/zika/reporting/2018-case-counts.html>; WHO, Available at: <https://www.who.int/emergencies/diseases/zika/india-november-2018/en/>), therefore, we propose that it is impractical to collect sewage samples for ZIKV detection from epidemic areas. Alternatively, we relied on the other suggestion from the reviewer to address this concern. We collected 10 liquid samples from cesspits at different locations. The characteristics of these cesspit samples are shown in Supplementary Table 2. Of these cesspits, the pH values range from 7.45-8.35; the chemical oxygen demand (COD) largely varied from 80-2300 mg/L; and the amount of ammonia nitrogen (NH₃-N) ranged from 25-430 mg/L (Supplementary Table 2). Subsequently, we tested the viral survivability in these cesspit samples. The viral survivability in the samples was measured by a time course plaque assay.

We analyzed 1×10^5 pfu/ml of initial viral titer at 28°C. One sample with the highest amount of COD and NH₃-N (Sample #10) absolutely inactivated ZIKV at the first time point of detection (1 hr), while other samples of the virus partially survived until 1 hr (Sample #9) or 8 hrs (Samples #7 and #8) post incubation. The virus survived up to 192 hrs in the rest of the samples (Figure 3a). Nonetheless, the previous literature indicates that the amount of infectious ZIKV particles discharged in patient urine might be 10-20 pfu/ml (Bonardo et al., 2016, PLoS Negl. Trop. Dis., 10, e0004816; Lustig et al., 2016, Euro. Surveill., 21, 30269). We therefore utilized 20 pfu/ml as the initial viral titer to further assess the ZIKV stability in these sewage samples. Consistently, the samples with high concentrations of COD and NH₃-N (Samples #7, #8, #9, #10) absolutely inactivated the viruses within a thirty minutes post incubation, while ZIKV maintained its survivability for more than 3 hrs in Samples #1 to #5 (Figure 3b). We proposed that the COD and NH₃-N concentrations in cesspit liquids may act as a key determinant for ZIKV survivability. These data indicate that ZIKV can survive in sewage under certain conditions. In the ZIKV epidemic area of America, the COD concentration of sewage varies from approximately 88 to 1323 mg/L, the pH ranges from 6.61 to 8.5, and the NH₃-N ranges from 5 to 192 mg/L, according to the literature of field surveillance (Supplementary Table 3). We therefore used sewage Sample-#5 (COD: 462 mg/L; NH₃-N: 85 mg/L; pH: 7.62) to mimic the sewage of the epidemic regions for mosquito breeding.

In the scenario of urination by an infected individual, the infectious urine might be discharged into a septic tank approximately every 2-3 hrs because a person normally urinates 6-8 times per day. To closely simulate the natural settings, we continuously added either 1/10 or 1/100 dilution of 20 pfu/ml ZIKV urine into the sewage sample

(Sample-#5) at 3 hrs intervals (Figure 3c). For continuous supplementation with a 1/10 dilution of 20 pfu/ml ZIKV urine, the viral titer was maintained at approximately 1-2 pfu/ml (Figure 3d). However, the viral titer was unable to be measured by a plaque assay at 1/100 dilution (the pfu was less than 1) (data not shown). We then reared the pupae of a field-derived *Aedes aegypti* (*A. aegypti*) strain, which was collected from the ZIKV epidemic area of Paraiba, Brazil, in these aquatic systems. The pupae emerged in the sewage successfully (Supplementary Figure 2b). The emerging adult mosquitoes were reared for 8 days before being subjected to ZIKV detection by RT-qPCR. Intriguingly, 1.9% (4/206) of *A. aegypti* adults that breeding in sewage with 1/10 dilution of 20 pfu/ml ZIKV urine (maintenance of 1-2 pfu/ml ZIKV), showed ZIKV RNA positivity. Nonetheless, none of the mosquitoes were ZIKV-positive by qPCR when breeding in 1/100 ZIKV urine dilution (Figure 3e). Consistently, one out of 78 mosquito pools (10 mosquitoes per pool) from the sewage with 1/10 dilution of 20 pfu/ml ZIKV urine was positive for the isolation of infectious ZIKV in Vero cells (Figure 3f). The infectious supernatant was further subjected to viral sequencing to confirm that the cytopathic effects (CPEs) were caused by ZIKV infection.

We next assessed whether the mosquitoes breeding in the infectious sewage could maintain ZIKV transmission cycle between hosts and mosquitoes. A "mosquito-mouse-mosquito" transmission model was previously used for a ZIKV transmission study (Liu et al., 2017, Nature, 545,482-486). The pupae of an *A. aegypti* Brazil Paraiba strain were reared in sewage maintained with 1-2 pfu/ml ZIKV. Eight days after adults emerged, thirty female *A. aegypti* mosquitoes were allowed to feed on a type I/II interferon receptor-deficient (*ifnagr^{-/-}*) C57BL/6 (AG6) mouse. Additionally, the infected mouse was subjected to feeding by naïve *A. aegypti* (Rockefeller strain) mosquitoes throughout the viremic stage. After 8 days, the fed mosquitoes were analyzed by RT-qPCR (Figure 3g). Notably, one out of the 18 mice exposed to infected mosquitoes was ZIKV-positive by either RT-qPCR (Figure 3h) or plaque assay (Figure 3i) from 1 to 7 days after mosquito feeding. The infected mouse died at 11 days after mosquito feeding. Intriguingly, among mosquitoes that fed on the viremic mouse (*A. aegypti*-BP-AG6-#26) from day 3 to day 5, 10%-100% mosquitoes successfully acquired ZIKV infection (Figure 3j). Therefore, the "host urination-mosquito breeding-naïve host" route may be a ZIKV transmission cycle that could contribute to ZIKV prevalence and circulation in nature.

Thus, we demonstrated that the field-derived *A. aegypti* mosquitoes from the ZIKV epidemic areas can acquire ZIKV infection during breeding in infectious sewage in natural settings and subsequently transmit the infection to hosts by biting. We have added the results and discussion to the revised manuscript (Line 233-265, Page 10; Line 339-360, Page 14).

#1C. However, an additional major concern is the assumption that significant amounts of infectious virus are shed with urine of infected patients. The authors cite references to support the notion that urine of ZIKV infected patients contains high viral loads –

however, these viral loads are based on RNA, not infectious virus. The statement in line 79-80 in which the authors say ‘...discharge high levels of infectious ZIKV (10^5 - 10^6 copies/mL)’ is highly misleading – the copy numbers are correct and have been reported, but again these are NOT necessarily infectious, and instead these ‘viral loads’ are based purely on viral RNA. The only study in which virus was titrated directly from a urine samples by plaque assay (reference 14) found only a single plaque in one sample and suggested an infectious viral titer of only 10 PFU/mL in the urine of this patient. All other tested urine samples were negative (or plaque titers were not determined). While other studies which isolated ZIKV from urine, may have involved samples with higher viral titers, these data were not provided in any of the cited publications. Thus, there is no convincing evidence for urine samples containing 10^3 PFU/mL or even 20 PFU/mL as used in some experiments. In cell culture, a few infectious virus particles may be sufficient to infect the culture and result in virus isolation. Without the acquisition of human urine samples of infected patients and titration of ZIKV from urine samples, this question will remain a major concern for the relevance of the presented study.

#1C Answer: Previous studies indicate that infectious ZIKV particles can be discharged by patient viruria. At least 5 ZIKV infectious strains were initially recovered and cultured from human urine samples (Oliveira, et al., 2018, Rev. Inst. Med. Trop. Sao Paulo, 60, e15; Zhang, Fu Chun, et al., 2016, Lancet Infect. Dis., 16, 641-642; Fonseca et al., 2014, Am. J. Trop. Med. Hyg., 91, 1035-1038; Hashimoto et al., 2017, Emerg. Infect. Dis., 23, 1223-1225; Bonaldo et al., 2016, PLoS Negl. Trop. Dis., 10, e4816). Additionally, results from two independent studies reported the infectious viral titer in the urine of ZIKV patients. Bonaldo and colleagues reported an infectious ZIKV titer of 10 pfu/ml in the urine of a patient (Bonaldo et al., 2016, PLoS Negl. Trop. Dis., 10, e4816). Lustig et al. reported that equivalent of 12-20 pfu/ml ZIKV was present in 3 patients from 5 to 26 days after the onset of Zika symptoms (Lustig et al., 2016, Euro. Surveill., 21, 30269). Therefore, we speculate that the rational amount of ZIKV may be 10-20 pfu/ml in the urine discharged by patients.

We therefore repeated the mosquito breeding studies in liquid conditions with 20 pfu/ml as a starting ZIKV titer. We first assessed ZIKV stability in human urine. To mimic the natural post-excretion conditions, we incubated ZIKV with 8 fresh human urine samples (PRVABC59 strain, final titer 20 pfu/ml). The mixture was then maintained at 28°C for 24 hrs. Interestingly, ZIKV survivability varied significantly among the individual urine samples. A titer of ZIKV was detected in 1 sample at 24 hrs. at 28°C using a plaque assay (Figure 1b). We next investigated whether the mosquitoes that were breeding in urine with infectious ZIKV might acquire ZIKV. Both *A. aegypti* and *Aedes albopictus* (*A. albopictus*) have been shown to carry and transmit ZIKV in several field studies. (Marchette et al., 1969, Am. J. Trop. Med. Hyg., 18, 411-415; Musso et al., 2016, Clin. Microbiol. Rev., 29, 487-524). We next investigated whether the different mosquito species breeding in human urine with ZIKV might acquire infection. In the first experiment, we incubated the pupae of either

A. aegypti (Rockefeller strain) or *A. albopictus* (Jiangsu strain) with the ZIKV-positive urine, in which the final ZIKV titers were 20 pfu/ml, 2 pfu/ml and 0.2 pfu/ml (Figure 1c). The pupae of these two species emerged in human urine (Supplementary Figure 2a). The adult mosquitoes were reared for 8 days and subjected to ZIKV detection by RT-qPCR. For the *A. aegypti* Rockefeller strain, 3.7% (8/215) and 2.7% (4/147) of mosquitoes in the adult stage, breeding in 20 pfu/ml and 2 pfu/ml ZIKV urine, respectively, were positive for ZIKV by RT-qPCR detection. However, none of mosquitoes (0/129) acquired the infection from human urine with 0.2 pfu/ml ZIKV (Figure 1d). Additionally, 2.7% (6/223) and 0.7% (1/149) of adult *A. albopictus* mosquitoes breeding in 20 pfu/ml and 2 pfu/ml ZIKV urine, respectively, showed positive detection of the ZIKV (Figure 1e). We next tested the infectivity of ZIKV virions in the mosquitoes breeding in ZIKV urine. In human urine with 20 pfu/ml ZIKV, two of 50 *A. aegypti* pools (Rockefeller strain, 10 mosquitoes per pool) and one of 93 *A. albopictus* pools (10 mosquitoes per pool) were positive for infectious ZIKV, as determined by ZIKV culture in Vero cells. However, under similar experimental conditions with 2 pfu/ml ZIKV, one of 65 *A. aegypti* pools was positive for the infectious virions, while no infection was detected from a total of 100 pools of *A. albopictus*. The infectious supernatant was further subjected to viral sequencing to confirm that the CPEs were caused by ZIKV infection (Figure 1f). Thus, we demonstrated that different *Aedes* mosquito species are susceptible to ZIKV infection from breeding in infectious aquatic systems.

We next assessed whether the mosquitoes breeding in ZIKV urine could transmit ZIKV to a naïve host. A "mosquito-mouse-mosquito" transmission model was previously used for the ZIKV transmission study (Liu et al., 2017, Nature, 545, 482-486). We first reared the pupae of the *A. aegypti* Rockefeller strain in urine with 20 pfu/ml ZIKV. Eight days after the adults emerged, twenty female *A. aegypti* mosquitoes were allowed to feed on a type I/II interferon receptor-deficient (*ifnagr^{-/-}*) C57BL/6 (AG6) mouse, an established animal model for ZIKV infection (Figure 2a) (Liu et al., 2017, Nature, 545, 482-486). Notably, three out of the 17 mice exposed to infected mosquitoes were ZIKV-positive by RT-qPCR detection at least one day post infection (Figure 2b). The infectious ZIKV particles were also detected at the viremic peak by a plaque assay (Figure 2c). We next validated this phenomenon with a field-derived *A. aegypti* strain, which was collected from the ZIKV epidemic area of Paraiba, Brazil. RT-qPCR detected that 2.3% (6/258) and 1.6% (3/191) of emerged adult mosquitoes (Brazil Paraiba strain), which were breeding in either 20 pfu/ml or 2 pfu/ml ZIKV urine, respectively, were positive for ZIKV (Figure 2d). Subsequently, the emerged female *A. aegypti* mosquitoes were fed on the AG6 mice, with the same experimental procedure as used for the Rockefeller strain. One out of 14 mice exposed to infected mosquitoes developed robust viremia from 1 to 7 days post feeding (Figure 2e and 2f). This infected mouse died 10 days post feeding, indicating efficient ZIKV transmission by the mosquitoes breeding in the ZIKV urine.

We have added the results and discussion to the revised manuscript (Line

#1D. *Finally, the modelling aspect of this study seems artificial and the estimated numbers of 12-16% of naïve mosquitoes acquiring ZIKV through contaminated urine highly unlikely based on the data presented (in which only few mosquitoes acquired ZIKV during development, see Fig1h). Without clear information on mosquito breeding in cesspits/septic tank in these locations, as well as actual data on infectious virus levels in the urine of ZIKV infected patients over time, these models cannot be relied upon. In addition, not enough evidence was given on the actual parameters used in the study (what numbers were used for e.g. extrinsic and intrinsic incubation periods), and the fact that sexual transmission was not included in the model will most likely confound results as well.*

#1D Answer: Thanks for the reviewer's question. We made an error in the result description in the last version of the manuscript. The correct statement should be "This model reveals that 5.5% to 17.2% of infected mosquitoes acquire the infection from human urine-contaminated water during the epidemic peak time (Moorea, $0.23\% / 4.15\% = 5.5\%$; Tahiti, $0.67\% / 3.90\% = 17.2\%$) (Figure 6d)". The ratio represents "the number of mosquitoes that become infected by sewage containing ZIKV / the total number of infected mosquitoes". The proportion of ZIKV-infected mosquitoes that breed in sewage containing ZIKV (the number of mosquitoes infected through sewage containing ZIKV / total number of mosquitoes) is estimated to be 0.23% to 0.67% during the epidemic peak times, which is consistent with our results in Figure 3f.

Here, we took advantage of the mathematical model to assess our assumption of ZIKV urine transmission in natural scenarios by combining our experimental results with the real datasets of incidence and seroprevalence. The parameter settings have been provided in the Methods section (Line 556-558, Page 22) and Supplementary Table 5. For the sexual transmission route, the estimated percentage of cases based on previous studies ranged from 3% (95% CI: 0.12%, 45.73%) (Gao et al., 2016, Sci. Rep., 6, 28070) to 23% (95% CI: 1%, 47%) (Towers et al., 2016, Epidemics, 17, 50-55). Indeed, the parameters from these 2 modeling settings share a large difference, indicating variations of sexual ZIKV transmission among human populations. We also include the sexual transmission into our model by applying the sexual parameter estimated by Gao et al., 2016. As the model framework shows that the sexual transmission will not influence the quantification of the urine transmission pathway.

#1E. *Another concern that is somewhere between major and minor is the evaluation of ZIKV positive/negative mosquitoes (Fig 1e/f). The presentation of ZIKV RNA relative to actin mRNA is not very useful for interpretation by the reader. Viral copy numbers estimated from a ZIKV standard curve would allow for much better evaluation of the data. Since ZIKV copy numbers were shown later on in Fig 2b/e, it is confusing why another way of presenting ZIKV RNA levels was shown here.*

*Furthermore, the numbers of mosquitoes that were in fact positive for infectious virus after exposure to ZIKV during development were very low (2/68 *A. aegypti* pools and 1/54 *A. albopictus* pools), suggesting that infection through larval water was in fact very inefficient. The minimum infection rate for *A. aegypti* was only 0.3% and the maximum infection rate 2.9%. However, this aspect is presented in a very misleading way throughout the manuscript. On a positive note, a low PFU/mL titer of 20 PFU/mL was used in these experiments, which may be slightly more relevant to field conditions than 1000 PFU/mL (although still, we have no evidence for any specific titer in sewage water).*

#1E Answer: Thank you for the reviewer's suggestion. For the experimental parameters in the revised manuscript, we have used 20 pfu/ml, 2 pfu/ml and 0.2 pfu/ml for the investigation. Please refer to response #1C.

In light of these data, we agree that the mosquito infective ratio after ZIKV exposure during breeding is low. Nonetheless, based on many field studies, the probability of arbovirus-carrying mosquitoes in the field may also be maintained at low levels, ranging from 0.03% to 0.88% in endemic areas (Chen et al., 2010, Vector Borne Zoonotic Dis., 10, 1017-1025; Tewari et al., 2004, Trop. Med. Int. Health, 9, 499-507; Dzul-Manzanilla et al., 2016, Trans. R. Soc. Trop. Med. Hyg., 110, 141-144; Ibanez-Bernal, 1997, Med. Vet. Entomol., 11, 305-309; Martinez et al., 2014, J. Am. Mosq. Control Assoc., 30, 143-146; Kabilan et al., 2004, Am. J. Infect. Control, 32, 391-396; Lutomiah et al., 2016, PLoS Negl. Trop. Dis., 10, e0004981; Ahmad et al., 1997, Southeast Asian J. Trop. Med. Public Health, 28, 138-142). The data indicate that the low infective ratios of mosquitoes derived from infectious sewage may reflect the real ZIKV prevalence in mosquitoes in nature.

#1F. *Some small grammar/English errors throughout the manuscript should be addressed (e.g. line 88-80: the use of 'Although' and 'however' is redundant; line 90: unsure what is meant by 'have habited')*

#1F Answer: We have revised the overall manuscript for language editing. The errors noted by the reviewer have been corrected (Line 99, Page 6; Line 101, Page 6).

#1G. *Figure 1a: The description makes this sound as if it's the experimental design, while really this is what is hypothesized to occur in nature, and what was mimicked in the lab. This is partially due to the use of past tense – the wording should be adjusted to improve clarity of the figure legend.*

#1G Answer: We have removed Figure 1a according to the reviewer's suggestion.

#1H. *Figure 1b/c: In these experiments, natural conditions of the bladder and natural habitats are being mimicked. Using 10^5 PFU/mL for the experiment in Fig 1b seems somewhat reasonable, since this much virus may be secreted into the bladder.*

However, the authors also used 10^5 PFU/mL for urine diluted 1:50 in water (Fig. 1c) and there is no evidence for any ZIKV patients shedding anywhere near 10^5 PFU/ML ZIKV in urine. The starting titer for this experiment should have been significantly lower.

#1H Answer: Since the rational ZIKV amount may be 10-20 pfu/ml in the urine discharged by patients, we used 20 pfu/ml ZIKV at 28°C as the starting titer for the investigation. Please refer to response #1C.

#1I. Supplementary 1: *Some of the text for reference 26 and 30 is cut off (size of table row too small?).*

#1I Answer: We have made changes in the Supplementary Table 1 according to the reviewer's suggestions.

Responses to Referee #2:

#2A. *In the proposed paper, the authors explore the extent to which ZIKV in urine can be transmitted to mosquitoes. They then incorporate urine driven mosquito ZIKV infection into mathematical models to help explain the epidemics in French Polynesia and Yap Island. That infectious ZIKV particles can exist in urine is known (Zhang et al., Bonaldo et al.,). That mosquitoes can get infected from urine is novel and interesting, however, it was unclear to me whether the concentrations used in the study were epidemiologically relevant. I also found the evidence for urine-mediated transmission being epidemiologically relevant to be weak. The authors state that 15% of mosquitoes in three outbreaks acquired infection from urine, however, this was made using simplistic models with no consideration of the epidemiological situation. Detailed comments below.*

#2A Answer: Thanks for the reviewer's general comments. We recognize that although our laboratory conditions do not completely represent the complex nature of human-virus-environment interactions, at a minimum provide proof-of-principle for ZIKV transmission via the urine-mosquito route. We have included new experimental data in our manuscript per the reviewers' suggestions.

The major changes are:

1. We demonstrated that different *Aedes* mosquito species are susceptible to ZIKV infection from breeding in human urine with lower ZIKV titers (20 pfu/ml and 2 pfu/ml) (Fig. 1d and e).

2. We found that either the *A. aegypti* Rockefeller strain or the field *A. aegypti* mosquitoes (Brazil Paraiba strain) breeding in human urine with a lower ZIKV titer (20pfu/ml) can transmit ZIKV to a naïve host (Fig. 2b-c and e-f).

3. We assessed the vector competence of different *Aedes* strains after feeding on a serial titration of ZIKV (Suppl. Fig. 4).

4. We demonstrated that the field-derived *A. aegypti* mosquitoes (Brazil Paraiba strain) from the ZIKV epidemic areas can acquire ZIKV infection during breeding in infectious sewage in natural settings and subsequently transmit the infection to hosts by biting (Fig. 3).

5. The parameters for mathematic modeling was optimized based on the data from the Brazil Paraiba strain to reflect the epidemiological relevance (Fig. 6a-d and suppl. Table 5).

Based on the reviewer's suggestions, we re-designed the experiments to closely simulate the natural settings. We hope that our additional experiments may substantially improve the revised manuscript. We have provided point-by-point responses to the questions and suggestions proposed by the reviewers.

#2B. *The authors use 10^3 PFU/ml for the initial concentration in urine, which seems high. Bonaldo et al (PLoS NTD 2016) found 10 pfu/ml. Lustig et al., had similar number (10-20 pfu/ml). Gourinat et al., tried but failed to isolate infectious particles from urine, suggesting that RNA but not whole virus was present. The 50-fold dilution used by the authors will still leave similar or even higher levels of virus than these other studies. The rational for the 1:50 dilution is unclear. The authors may want to consider the effect of more diluted urine samples. Given that the proportion of infected mosquitoes was already very low (~1% in 1:50 dilution versus 4% in undiluted), there may no longer be transmission. This is an important consideration as if urine to mosquito transmission requires high concentrations, this is unlikely to be epidemiologically relevant.*

#2B Answer: We agree with the reviewer's concerns. We have changed the experimental parameters to lower ZIKV titers, which reflect the epidemiological relevance much better. Please refer to response #1C.

#2C. *The mosquitoes in the island outbreaks considered in the models were not the ones considered in their experiments. In French Polynesia, aedes polynesiensis mosquitoes appeared to have an important role. In Yap, Aedes Hensili seemed key with very little aegypti present (see Ledermann, PLoS NTD, 2014). The relevance of the experimental data in describing these outbreaks is therefore unclear. It is also unclear which species the fixed parameters in the model are fit to (including mosquito lifespan, extrinsic incubation period etc).*

#2C Answer: Thank you for the reviewer's question. The informative prior distributions were assumed for the mosquito lifespan, the duration of the infectious period, and the extrinsic incubation period based on *Aedes* mosquitoes (mostly *A. aegypti* and *A. albopictus*) as noted in the literature. Indeed, the ability of *A. aegypti* to transmit ZIKV in French Polynesia has been demonstrated; however, previous studies indicated that *A. polynesiensis* exhibited poor vector competence for ZIKV (Richard et al., 2016, PLoS Negl. Trop. Dis., 10, e0005024; Musso et al., 2018, Lancet Infect. Dis.,

18, e172-e182). On the Yap Island, *A. hensilli* was considered the main vector of the ZIKV outbreak in 2007. However, there are no available experimental data on *A. hensilli* yet, and we found it impossible to obtain this species in China. We therefore decided to remove the results of the simulation and estimation from Yap Island.

In the revised manuscript, we used a field-derived *A. aegypti* strain from the ZIKV epidemic area in Brazil (Brazil Paraiba strain) to validate the phenomenon. We have applied the data from the Brazil Paraiba strain to fit the model settings. The modeling parameters are provided in Supplementary Table 5.

#2D. *A further key issue with the model is that there is no assessment of whether the models that include urine-mediated transmission outperform models that do not include it. ϕ , the water contamination rate is very close to 0 (0.0003) indicating that W is very small which suggests there may be little differences between models that do and do not include a urine component, however, this should be assessed.*

#2D Answer: As shown in equation [8], the contaminated proportion of water patches (ranging from 0 to 1) is influenced by the product of H_i (number of Zika patients) and ϕ (water contamination rate by the urine from a fresh Zika patient per day).

$$\frac{dW}{dt} = P_{pH} \phi H_i (1 - W) - \delta W \quad [8]$$

Our objective was not to improve the predictive power but to assess our hypothesis of urine transmission and to test some scenarios with parameters in the acceptable ranges for reducing associated uncertainties.

#2E. *There is no information on the priors used – especially on the key ϵ and ϕ parameters. For some of the parameters in the original model that the authors rely on, the posterior is highly dependent on the prior (see Figures 10-13 in Champagne et al., eLife) – it wouldn't be surprising if the priors had an important role with these two additional parameters too.*

#2E Answer: Informative prior distributions were assumed for the mosquito lifespan, the extrinsic incubation period in mosquitoes, the infectious period in humans, the intrinsic incubation periods, and the survivability of ZIKV in human urine. The range for the prior initial numbers of infected / exposed individuals and mosquitoes was obtained from the previous report (Champagne et al., 2016, Elife, 5, e19874). The prior information on ϵ and ϕ has been added as suggested (Supplementary Table 5).

#2F. *It is unclear which model the authors use – the original papers presented a number of competing models.*

#2F Answer: Figure 6a is a schematic representation of the model derived by Pandey et al. (Pandey et al., 2013, *Math Biosci*, 246, 252-259) and explicitly formulates the mosquito population. We have added a related description to the Methods section (Line 520-521, Page 21).

#2G. *The model uses a vertical transmission rate of 0.25 based on a reference. However in the reference, the authors estimate that 1:290 vertical transmission proportion for aegypti and 0 for albopictus. The authors should clarify how they obtained their estimate. Again, values for polynesiensis/hensilli are not available.*

#2G Answer: Thank you for this question. A ratio of 1:290 was provided in the literature (Thangamani et al., 2016, *Am. J. Trop. Med. Hyg.*, 95, 1169-1173). There are no available experimental data on *A. hensilli* and *A. polynesiensis*, and we found it impossible to obtain and rear these species in China. We are sorry that we cannot obtain the values for *A. polynesiensis* and *A. hensilli* in our lab settings.

#2H. *In a number of places in the introduction, the authors state that ZIKV develops a high viremia. However, compared to CHIKV and DENV, viremia in ZIKV patients seems generally low (1-2 logs lower) (see e.g., Waggoner et al., CID 2016).*

#2H Answer: We agree with the reviewer's concern. Indeed, clinical surveillance showed a relatively low ZIKV viremia (10^2 - 10^3 pfu/ml) in serum samples (Gourinat et al., 2015, *Emerg. Infect. Dis.*, 21, 84-86; Barzon et al., 2016, *Euro. Surveill.* 21, 30159; Waggoner et al., 2016, *Clin. Infect. Dis.*, 63, 1584-1590), compared to that of CHIKV and DENV. Therefore, we assessed the vector competence of different *Aedes* strains after feeding on a serial titration of ZIKV (PRVABC59 strain). Intriguingly, the acquisition of human blood with 1×10^5 pfu/ml, but not 1×10^3 pfu/ml ZIKV, resulted in infection and transmission by the *A. aegypti* Rockefeller strain, the *A. albopictus* Jiangsu strain, and the *A. aegypti* Brazil Paraiba strain (Supplementary Figure 4). In addition, emerging evidence indicates that low-passage field mosquitoes, such as *A. aegypti* and *A. albopictus* collected from the Americas, have an unexpectedly low vector competence for ZIKV transmission by oral blood feeding (Chouin-Carneiro, et al., 2016, *PLoS Negl. Trop. Dis.*, 10, e0004543), suggesting that an alternative route might contribute to the prevalence of mosquito infection in nature.

Here, we report that the mosquitoes breeding in a cesspit water system contaminated by human urine containing ZIKV can acquire and transmit ZIKV. Our study suggests that the mosquitoes could acquire ZIKV not only through feeding on a viremic host but also through breeding in urine excreted by ZIKV patients, suggesting an alternative "host urination-mosquito breeding" transmission cycle that may exist for ZIKV transmission between mosquitoes and hosts in nature. We have corrected the "high viremia" in the introduction (Line 71 and Line 74, Page 4). The results and discussion have been added to the revised manuscript (Line 188-201, Page 9; Line

#2I. *The authors state that “consistently, the two field mosquitoes all acquired ZIKV infection during incubation” (line 155) – this is misleading as written as only a very small proportion become infected.*

#2I Answer: We have removed this sentence according to the reviewer's suggestion.

Responses to Referee #3:

#3A. *This manuscript by Liu et al. provides experimental evidence that Zika virus-positive urine could serve as a source of virus for acquisition by mosquitoes. In this study, the authors determined the stability of Zika virus in different urine samples and under different environmental (but laboratory) conditions. The authors then reared *Aedes aegypti*, *Ae. albopictus*, and *Culex quinquefasciatus* in water spiked with ZIKV-infected urine and assayed for infection via QRT-PCR and cell culture. In addition, mosquitoes that were reared in urine water were allowed to feed on Type I and II, IFN-deficient mice and transmission of Zika virus was demonstrated, implicating ZIKV-positive urine as a previously unidentified source for mosquito infection. While these results are interesting and worthy of follow up, enthusiasm for the conclusions drawn from this work are diminished by the experimental set up, which is nowhere close to biologically relevant both in terms of the volume of water that mosquitoes were reared in as well as the complete absence of other organic material that might be present in waste water that might also inactivate Zika virus (or influence pH)*

#3A Answer: We agree with the reviewer's concerns. We have changed the experimental parameters to better reflect the epidemiological relevance. Please refer to response #1B.

#3B. *appropriate controls demonstrating the peroral vector competence of their mosquito strains for ZIKV are missing, i.e., it is important to know what the infection, dissemination, and transmission rates of these mosquitoes are when exposed to a ZIKV-positive bloodmeal.*

#3B Answer: According to the reviewer's suggestion, we assessed the vector competence of different *Aedes* strains after feeding mosquitoes on a serial titration of ZIKV (PRVABC59 strain). The mosquitoes were fed with human blood (50% v/v) and a supernatant from ZIKV-infected Vero cells (50% v/v). The ZIKV load was determined from the midguts, the heads and the salivary glands at a time course. The infection, dissemination, and transmission rates of these mosquitoes were subsequently calculated accordingly. Intriguingly, acquisition of human blood with 1×10^5 pfu/ml, but not 1×10^3 - 1×10^4 pfu/ml ZIKV, resulted in efficient infection and transmission by the A.

aegypti Rockefeller stain, the *A. albopictus* Jiangsu strain, and the *A. aegypti* Brazil Paraiba strain (Supplementary Figure 4). However, based on the low viremia level reported in the published literature (Waggoner et al., 2016, Clin. Infect. Dis., 63, 1584-1590; Lustig et al., 2016, Euro. Surveill., 21, 30269), we speculate that the mosquitoes may exhibit a low vector competence for ZIKV by oral blood feeding, suggesting that an alternative route might contribute to the prevalence of mosquito ZIKV infection in nature. We have added the results to the revised manuscript (Line 188-201, Page 9).

#3C. *Furthermore, the Aedes aegypti strains tested were the Rockefeller strain, which is highly lab adapted and two Asian strains. Critically, there were no American Aedes aegypti included in the study.*

#3C Answer: We agree with the reviewer's concern. We therefore used a field-derived *A. aegypti* strain from the ZIKV epidemic area in Brazil (Brazil Paraiba strain) for the investigation. Please refer to the data in response #1B.

#3D. *Finally, the model does not provide convincing evidence that ZIKV-positive urine is a source of ZIKV infection for mosquitoes.*

#3D Answer: In this study, we proposed an alternative ZIKV transmission model with host urination and mosquito breeding in an aquatic environment. The transmission steps include: (1) a Zika patient may release 1,000-2,000 ml urine daily, resulting in infectious ZIKV particles excreted by patient urination into natural environments; (2) the mosquitoes breeding in the cesspit/sewage contaminated by ZIKV may acquire the infection; and (3) the infected adult mosquitoes transmit the virus to hosts. We believe we have provided evidence in the previous three steps.

Regarding the viruria by the infected host (corresponding to Step 1), the literature indicates that infectious ZIKV particles can be discharged in the patient's urine. At least 5 infectious ZIKV strains were initially recovered and cultured from human urine samples (Oliveira, et al., 2018, Rev. Inst. Med. Trop. Sao Paulo, 60, e15; Zhang, Fu Chun, et al., 2016, Lancet Infect. Dis., 16, 641-642; Fonseca et al., 2014, Am. J. Trop. Med. Hyg., 91, 1035-1038; Hashimoto et al., 2017, Emerg. Infect. Dis., 23, 1223-1225; Bonaldo et al., 2016, PLoS Negl. Trop. Dis., 10, e0004816). Additionally, two independent studies reported infectious viral titers in the urine of ZIKV patients. Bonaldo and colleagues reported an infectious ZIKV titer of 10 pfu/ml in the urine of a patient (Bonaldo et al., 2016, PLoS Negl. Trop. Dis., 10, e0004816). Lustig et al. reported that the equivalent of 12-20 pfu/ml ZIKV was present in 3 patients from 5 to 26 days after the onset of Zika symptoms (Lustig et al., 2016, Euro. Surveill., 21, 30269). Therefore, the rational ZIKV amount may be 10-20 pfu/ml in the urine discharged by patients.

Regarding ZIKV acquisition by breeding in the cesspit/sewage environment

(corresponding to Step 2), new evidence in the revised manuscript suggests that the field-derived *A. aegypti* mosquitoes from the ZIKV epidemic areas (Brazil Paraiba strain) can acquire ZIKV infection by breeding in infectious sewage with 1-2 pfu/ml ZIKV. Please refer to response # 1B.

For ZIKV transmission by the cesspit-derived infected mosquitoes (corresponding to Step 3), we used a "mosquito-AG6 mouse-mosquito" model for the ZIKV transmission study (Liu et al., 2017, Nature, 545, 482-486). The *A. aegypti* pupae were reared in the infectious sewage liquid (maintained at 1-2 pfu/ml ZIKV). The resulting infected mosquitoes then transmitted the infection to hosts by feeding. Please refer to response # 1B.

Overall, we believe that the aforementioned results and literature have provided pieces of evidence that ZIKV-positive urine is a source of ZIKV infection for mosquitoes. We have added this discussion to the revised manuscript (Line 339- 346, Page 14).

#3E. *Grammar and syntax need to be improved throughout.*

#3E Answer: Thanks for the reviewer's suggestion. We have overall revised the manuscript by language editing.

#3F. *Line 44 and 64, stating that ZIKV is maintained between mosquitoes and primates is misleading. It still is not clear how ZIKV is maintained in sylvatic transmission cycles, evidence suggests nonhuman primate involvement but it is not definitive.*

#3F Answer: We have corrected this sentence according to the reviewer's suggestion (Line 45, Page 3; Line 70, Page 4).

#3G. *Line 45, clinical studies have shown that ZIKV RNA persists in urine for an extended period of time. It is less clear how long infectious virus is shed in urine. The use of 'living infectious virus' is inaccurate, please revise.*

#3G Answer: Two independent studies have reported infectious viral titers in the urine of ZIKV patients. Bonaldo and colleagues reported an infectious ZIKV titer of 10 pfu/ml in the urine of a patient 5 days after the disease onset. (Bonaldo et al., 2016, PLoS Negl. Trop. Dis, 10, e0004816). Lustig et al. reported that the equivalent of 12-20 pfu/ml ZIKV was present in 3 patients from 5 to 26 days after Zika symptom onset (Lustig et al., 2016, Euro. surveill., 21, 30269). Altogether, the viremia discharged by patients may be present for weeks after the infection. Nonetheless, we agree with the reviewer's concern that the time-span of infectious ZIKV shedding by patients is still largely unclear. We have revised the manuscript according to the reviewer's suggestion (Line 46, Page 3).

#3H. *Line 66, please include references to confirm that ZIKV develops high viremia in human blood. Most studies measure vRNA loads in human samples.*

#3H Answer: Thanks for the reviewer's suggestion. Please refer to the Answer to #2H.

#3I. *Line 76 replace "living" with "infectious".*

#3I Answer: We have made the change according to the reviewer's suggestion (Line 82, Page 4).

#3J. *Line 79, the authors claim that high levels of infectious ZIKV are discharged in the urine but then list a range of vRNA copies not PFU. How much infectious ZIKV is discharged with urine, it is likely at least 1000-fold lower than the amount of vRNA detected. Again, it is disingenuous to state that viremia persists for a long time. vRNA can persist for extended periods in urine but how long is it infectious? Also, for ZIKV shed in semen it has been shown that the PFU:particle rate fluctuates drastically over time. Does this also happen with urine?*

#3J Answer: We fully agree with the reviewer's concern. There is no accurate data available for the persistence of ZIKV infectious particles in patient urine. In addition, some studies have reported 10-20 pfu/ml ZIKV discharged in patient urine at various time points after the disease onset (Bonaldo et al., 2016, PLoS Negl. Trop. Dis., 10, e0004816; Lustig et al., 2016, Euro. Surveill., 21, 30269). Therefore, we changed our experimental parameters to better reflect the epidemiological relevance. Please refer to response #1C.

#3K. *Line 78, If a single human is shedding ZIKV in urine into gallons of waste water will the virus be diluted out to the point it won't matter? Given the variability reported later in the study about the heterogeneity of ZIKV stability in urine samples from different individuals, might ZIKV not be inactivated once introduced into the milieu of urine samples present in waste water?*

#3K Answer: The current results indicated that the maintenance of 1-2 pfu/ml ZIKV in sewage, but not a dilution of 0.2 pfu/ml, is able to cause the infection of adult mosquitoes through breeding. Furthermore, the infected mosquitoes derived from the infectious sewage can transmit the virus to a host through a blood meal (Figure 3h and 3i). We therefore speculate that the infectious urine shed by a few patients into a large collection of water (a very high dilution) may not be sufficient for the ZIKV infection of mosquitoes by breeding. Nonetheless, in scenarios during disease outbreaks, many infected individuals (with or without symptoms) may intensively discharge a large volume of infectious urine into a restricted aquatic habitat of mosquitoes. We therefore speculate a scenario during a Zika outbreak in which the

accumulating ZIKV infectious particles discharged by a large number of patients might be sufficient to facilitate mosquito infection by breeding in cesspit conditions, thereby establishing the "host urination-mosquito breeding" transmission cycle.

#3L. *Line 87-96 The scientific premise is faulty. Mosquito productivity depends on myriad factors, including the nutritional quality of the larval environment, container type, surrounding environmental conditions, etc. While it is generally accepted that A. aegypti prefer to oviposit in clean water, there have been documented cases of immature development in polluted water and raw sewage as demonstrated by Supplementary Data 1. But it is not clear how epidemiologically important these types of containers are for oviposition for A. aegypti or Ae. albopictus. It would be a potentially very different situation if the primary vector for ZIKV were, for example, Armigeres subalbatus, which is known to breed in sewage. However, this mosquito species is not found in the New World.*

#3L Answer: We understand the reviewer's concern. A common concept is that *Aedes* mosquitoes mainly breed from clean water. However, accumulating evidence indicates that both *A. aegypti* and *A. albopictus* have evolved to oviposit and breed in wastewater with low oxygen and high turbidity, such as cesspits, septic tanks and sewers (Supplementary Table 1). Notably, the population of adult *A. aegypti* mosquitoes emerging from waste water may be equal to or even larger than those emerging from traditional breeding sites (clean water) in Brazil (Herman et al., 2015, Communication 6, 73-80) and Puerto Rico (Burke et al., 2010, Med. Vet. Entomol. 24, 117-123).

We agree that evidence is still not clear regarding the epidemiological importance of cesspit breeding for ZIKV transmission. Further field surveillance is definitely needed to investigate the breeding behavior of these *Aedes spp.* in cesspit conditions. Nonetheless, given the strong evidence for the wide existence of sewage-derived *Aedes* productivity (Supplementary Table 1), it is rational to consider potential routes of ZIKV transmission by these sewage-derived mosquitoes that may acquire the infection by either blood feeding or water breeding.

#3M. *Line 113-127, what volume of water were the aquatic stages reared in and what was the overall infectious ZIKV titer in this volume of water and how does this relate to what one might expect to find in a septic tank, for example? Really, what I am asking is how biologically relevant is this in terms of a route of transmission to mosquito larvae or pupae? Did you try to estimate the minimum infectious dose for the larvae and pupae? Is this really the best experiment? It certainly would have been more stringent to include additional ZIKV-negative urine samples mixed in as a follow up as well as other biologic material that might impact ZIKV infectivity.*

#3M Answer: The aim of the experiment in Supplementary Figure 1 was merely used to assess the susceptibility of larvae and pupae to ZIKV infection. We concluded that

the pupae were the most susceptible stage, therefore using the pupae for further investigation. Moreover, we have changed the experimental parameters to better reflect the epidemiological relevance. Please refer to responses #1B and 1C.

#3N. *Line 125, it is surprising that the pupal stage was the most susceptible. What is the proposed explanation since pupae do not eat and breathe through respiratory trumpets?*

#3N Answer: Thank you for the reviewer's question. Indeed, the aquatic stages, including the pupal stage, are susceptible to infection by some viruses. For example, the *Aedes* densovirus can infect the mosquito larvae and pupae in the aquatic environment, and the anal papillae is the most commonly infected organ (Afanasiev et al., 1999, *Virology*, 257, 62-72; Buchatsky, 1989, *Diseases of aquatic organisms*, 6, 145-150.). We therefore speculated that the pupae may get ZIKV infection by the anal papillae or cavities of other organs, but not through the intestinal tract or respiratory trumpets, in the water system.

#3O. *Line 129-149 and 168-188, How were the adult mosquitoes processed or handled to eliminate the possibility of surface contamination? While I acknowledge that surface contamination is unlikely given the time frame of testing adults after emergence, testing for infectious ZIKV in pools of adults indicate a very low infection rate (~0.2%) suggesting that QRT-PCR is detecting residual vRNA that is not actually infectious. And how do you reconcile this very low infection rate in your pools with a fairly moderate transmission rate to mice? Again, I question the biological relevance. The mouse model used is very similar to the AG129 mouse model in which it has been shown that less than 1 PFU can cause high viremia and lethal disease in that model (see Salvo et al. 2018 PLoS NTD). It remains unclear how much ZIKV needs to be delivered by a feeding/probing mosquito to initiate infection in a human. Finally, all of the mouse serum was screened via QRT-PCR. It would be informative to verify that there is actually replicating virus in these animals.*

#3O Answer: We agree with the reviewer's concerns. Adult mosquitoes were analyzed for viral detection 8 days after pupal eclosion. Given the high sensitivity, the detection by qPCR amplifies viral genomic fragments, which may be from either the inactive virus in the mosquitoes or surface contamination in the aquatic stage. We agree that qPCR detection cannot fully reflect ZIKV infectivity in mosquitoes. To address this concern, we isolated infectious ZIKV particles in mosquitoes breeding from infectious human urine and sewage. The data are shown in the Figure 1f and 3f.

Based on the reviewer's suggestion, we assessed whether the mosquitoes breeding in the infectious sewage maintained the ZIKV transmission cycle between hosts and mosquitoes. A "mosquito-mouse-mosquito" transmission model was previously used for a ZIKV transmission study (Liu et al., 2017, *Nature*, 545, 482-486). The pupae of an *A. aegypti* Brazil Paraiba strain were reared in sewage maintained

with 1-2 pfu/ml ZIKV. Eight days after adults emerged, thirty female *A. aegypti* mosquitoes were allowed to feed on a type I/II interferon receptor-deficient (*ifnagr*^{-/-}) C57BL/6 (AG6) mouse. After the initial feeding, the infected mouse was subjected to feeding by naïve *A. aegypti* (Rockefeller strain) throughout the viremic stage (Figure 3g). Notably, one out of 18 mice exposed to infected mosquitoes was ZIKV-positive by either RT-qPCR (Figure 3h) or plaque assay (Figure 3i) 1 to 7 days post feeding. The infected mouse died at 11 days after exposure to infected mosquitoes. The blood fed mosquitoes were reared for an additional 8 days for RT-qPCR detection (Figure 3g). Intriguingly, among mosquitoes that fed on the viremic mouse (*A. aegypti*-BP-AG6-#26) from day 3 to day 5, 10%-100% of mosquitoes successfully acquired ZIKV infection (Figure 3j). Therefore, the "host urination-mosquito breeding-naïve host" route may be a ZIKV transmission cycle that contributes to ZIKV prevalence and circulation in nature. We have added these results to the revised manuscript (Line 250-265, Page 11).

#3P. *Line 161, were the 2 DENV-2 positive mosquitoes positive by QRT-PCR or via some other method to detect infectious virus?*

#3P Answer: We tried to isolate infective DENV virions from the mosquitoes (*A. aegypti* Rockefeller strain) breeding in the urine (Donor 3) containing DENV2 (1×10^3 pfu/ml), with the same experimental procedure as in Figure 1f. However, no positive samples were identified from 50 mosquito pools (10 mosquitoes per pool). We have added the results to the revised manuscript (Line 180-182, Page 8).

#3Q. *Line 192, correlated should be positively (or negatively) correlated measured by a statistical assay.*

#3Q Answer: We have made the correction according to the reviewer's suggestion. We noted that the pH values of human urine were positively correlated with ZIKV survivability. Linear regression analysis was used to assess the correlation between ZIKV survivability and the urine pH value in human urine ($r = 0.8214$, $P < 0.0001$). (Line 276, Page 12).

#3R. *Line 228, it is unclear whether the model is parameterized correctly. The most plausible vector for ZIKV during the Yap outbreak was Aedes hensilli not Aedes aegypti, so the findings of this manuscript are not directly relevant. I don't believe Aedes aegypti have been found on Yap. There are two potential vectors in French Polynesia: Aedes aegypti and Aedes polynesiensis. During the 2013 outbreak over 2000 female Ae. Aegypti were collected and tested for ZIKV and only a single pool was positive. Likewise, what are the sanitation systems like on these islands and are they likely to produce large numbers of mosquitoes?*

#3R Answer: Thanks for the question. *A. hensilli* was considered the main vector of the ZIKV outbreak on Yap Island in 2007. However, very small proportion of *A. aegypti*

were sampled during field surveillance in 2007 (Ledermann et al., 2014, PLoS Negl. Trop. Dis., 8, e3188). Since there is no available experimental data on *A. hensilli* and it was impossible to obtain this species in China, we decided to remove the results of simulation and estimation from Yap Island. The ability of *A. aegypti* to transmit ZIKV in French Polynesia has been demonstrated; however, the previous studies indicated that *A. polynesiensis* has a poor vector competence for ZIKV (Richard et al., 2016, PLoS Negl. Trop. Dis., 10, e0005024; Musso et al., 2018, Lancet Infect. Dis., 18, e172-e182).

Regarding the sanitation systems, we have provided the surveyed household data from the epidemic areas (Figure R1). Of 170 randomly surveyed households, 1366 water holding habitats were identified. Larvae and/or pupae were collected from 586 of these containers and 85% of surveyed households had at least one infested habitat (Ledermann et al., 2014, PLoS Negl. Trop. Dis., 2014, 8, e3188). Since the local households lack a sanitary sewage treatment system, the risk of human urine directly contaminating the aquatic habitat where mosquitoes breed is high.

Figure R1. Typical water holding containers at individual homes including water barrels, coconut shells and cooking utensils (from Ledermann et al., 2014, PLoS Negl Trop Dis. 2014, 8, e3188).

#3S. Line 244, the use of 'life cycle' is incorrect, it should be transmission cycle.

#3S Answer: We have made the change according to the reviewer's suggestion (Line 382, Page 15).

Reviewers' Comments:

Reviewer #1:

Remarks to the Author:

After review of the revised manuscript 'Viruria facilitates Zika virus prevalence and transmission by mosquitoes' by Liu et al., I can conclude that many of my concerns were addressed appropriately. The authors have added important additional data to improve the manuscript.

Briefly, I particularly appreciate the following additions:

- 1) The authors used lower Zika titers for the initial experiments shown in Figure 1. The authors show that even with a final Zika virus titer of 2 PFU/mL some pupae will become infected
- 2) New data to mimic conditions in sewage settings such as cesspits in the new Figure 3.
- 3) The authors have quantified their viral copy numbers directly instead of using a housekeeping gene ratio, which allows better interpretation of data

My main concern is still related to the strong conclusions drawn from the data. The title in itself implies that viruria is important for Zika virus prevalence and transmission during outbreaks – the authors have NOT shown this! The authors have shown that this may be a possible scenario that should be studied in the field. However, these laboratory generated data cannot conclude that this happens in the field – even the best model remains a model, especially since it is impossible to conclude whether the conditions mimicked in the lab truly represent natural scenarios. While I appreciate Figure 3, only 3 out of the 10 tested sewage samples allowed Zika to remain infectious by 8h. Further the addition of 20 PFU/mL ZIKV every 3hrs to the sewage (as used in Figure 3c/d), is also just based on guesswork. Yes, infectious ZIKV has been found in urine and 20 PFU/mL may be a reasonable titer, but we do not know if every urine sample throughout the day would contain infectious Zika. Based on how rarely it is found, this seems unlikely.

These concerns are not meant to prevent publication of the data, which are interesting. I would simply urge the authors to change their title, highlight the limitations of the study a bit more in the discussion, and instead highlight the main conclusion that can be drawn – that low levels of infectious Zika virus present in water during development can infect mosquitoes during development and result in mosquitoes capable of transmission.

Remaining minor concerns:

Overall, there are still numerous English language errors and grammar issues throughout the manuscript. I urge the authors to proof-read and correct these. In some cases these affect clarity, e.g. lines 802-803 'The data for the upper mosquito number are represented as the percentage of infection' – this sentence makes little sense to me. Other grammar errors occur but are not limited to lines 101, 104, 121, and others including figure legends.

It is not always clear what the 'mock' control is that the authors are showing. In Figure 3d for example, was the mock control just incubated in water? Or kept in urine? Similarly, the authors should check all figure legends and explain what the mock controls represent, where necessary.

The authors mention in response #1B that 'the infectious supernatant was further subjected to viral sequencing to confirm that the cytopathic effects are caused by ZIKV infection'. However, I did not find this information in the methods or main text. I think it would be valuable to add that this was done to the manuscript since it is a concern when using unclean samples such as these for plaque assay (e.g. adding it to the methods after line 476).

Consistent use of a log scale on y-axes would be preferred to switching back and forth (e.g. Figure 3h-j).

Line 356-359: 'with the accumulating ZIKV infectious particles discharged by a large number of patients'. The authors should keep in mind that for each infected patient that may have infectious ZIKV in their urine, there are probably multiple uninfected people (and patients without infectious ZIKV in their urine) discharging urine that would dilute the virus. This statement thus seems a bit grand and not entirely warranted.

Reviewer #2:

Remarks to the Author:

The authors have done an impressive amount of additional work to attempt to address the reviewer concerns. In particular, the use of sewage and lower viral concentrations are important additions. However, I am still far from convinced as to the epidemiological relevance of this observation. The additional experiments have highlighted that ZIKV doesn't survive very long in realistic conditions (Figure 3b). Importantly, readers are likely to be focused on the finding in the abstract that 5-17% of infections at the peak of the epidemics in French Polynesia came from urine infections – however, the evidence for this is very weak. I would strongly urge the authors to concentrate the article on the observation that under certain conditions, ZIKV can be transmitted from urine to mosquitoes and not make broad conclusions of the epidemiological relevance.

The major problem with the model is with the epsilon estimate. Epsilon represents the critical probability of ZIKV transmission during mosquito breeding in contaminated water. In Moorea, the authors provide an estimate of 0.49 with a standard deviation of 0.24, which will represent 95% confidence intervals of the entire range of probabilities (i.e. 0 to 1) – i.e., there is no information. The estimates in Tahiti are similar. In their own experiments, the authors are only able to infect a tiny proportion of mosquitoes in carefully selected conditions (i.e., only in a subset of sewage samples and urine samples could they get any infections) – therefore a probability of infection of 50% makes no sense. This parameter determines the transitions from the sewage compartment to the mosquito compartment so the overall proportion of infections that come from sewage is completely sensitive to this parameter.

In the recent experiments, the authors use sewage samples. They previously demonstrated that in water, ZIKV survived up to 192 hours. With the sewage samples, they appeared to have stopped the experiments after 8 hours (at which point the pfu had dropped a lot and was only detectable at all in 4 samples, Figure 3b). Even if it shows a negative result – the authors should present a longer time-frame of the survivability in these more realistic conditions.

Relatedly, in the new experiments, the authors also stopped the experiments after 24 hours when looking at survivability in urine at 28C (at which point it was only detectable in a single sample, Figure 1b) – again even if it shows a negative result, the authors should extend this analysis so that it is consistent with that in Fig 1a.

28C is also a high water temperature to be using as a realistic temperature for urine in sewage systems, even in tropical settings where ZIKV has been observed. In particular, nighttime temperatures will be important and likely to be substantially lower. At the very least, this weakness should be discussed.

There remain a lot of grammatical errors – I found the paper difficult to follow in a number of places.

Reviewer #3:

Remarks to the Author:

This iteration of the manuscript by Liu et al. is quite responsive to reviewer concerns and includes significant additions in experimental data. While these results are interesting and much improved in terms of biological relevance, I still have several concerns.

1.) Similar to Reviewer 1, I believe that the concept that mosquitoes can acquire ZIKV from sewage water is interesting and worthy of publication on its own without the grandiose conclusions about contributions to transmission. It is fine to speculate about some of this but in so doing it is necessary for the authors to discuss potential limitations to their experimental design and acknowledge the need for epidemiological follow up.

2.) For Line 197-201, I agree that low viremia has been reported in the literature but I disagree with the speculation that mosquitoes may have a low overall vector competence for ZIKV based on these data and the author's demonstration of poor infection rates for mosquitoes exposed to bloodmeal titers $<10^5$ PFU/ml. 1.) Viremia titers reported in the literature from infected humans are difficult to interpret because they are a snapshot in time and in all likelihood the result of an individual seeking treatment after symptom onset. As a result, it is possible that these titers may be past peak titer in that individual. 2.) Membrane feeding can influence the magnitude of the observed effect as compared to feeding on a viremic host, i.e., mosquitoes that are allowed to feed on viremic hosts (in general) have higher infection, dissemination, and transmission rates as compared to mosquitoes exposed to comparable bloodmeal titers via membrane feeding systems.

3.) I still continue to question the relevance of the mouse transmission model utilized here. At the very least there should be some discussion about the overall susceptibility of this model to Zika virus infection as I indicated in my previous review. This may help reconcile the very low infection rate detected when mosquito pools are screened vs. the low-to-moderate transmission rate to mice.

4.) I still question the veracity of the model and am concerned about potential misinterpretation.

Responses to Referee #1:

#1A. *My main concern is still related to the strong conclusions drawn from the data. The title in itself implies that viraemia is important for Zika virus prevalence and transmission during outbreaks – the authors have NOT shown this! The authors have shown that this may be a possible scenario that should be studied in the field. However, these laboratory generated data cannot conclude that this happens in the field – even the best model remains a model, especially since it is impossible to conclude whether the conditions mimicked in the lab truly represent natural scenarios. While I appreciate Figure 3, only 3 out of the 10 tested sewage samples allowed Zika to remain infectious by 8h. Further the addition of 20 PFU/mL ZIKV every 3hrs to the sewage (as used in Figure 3c/d), is also just based on guesswork. Yes, infectious ZIKV has been found in urine and 20 PFU/mL may be a reasonable titer, but we do not know if every urine sample throughout the day would contain infectious Zika. Based on how rarely it is found, this seems unlikely.*

#1A Answer: We agree with the reviewer's concern. According to the suggestions, we have changed the title to "Zika viraemia causes infection in mosquito by breeding in infectious aquatic environments" to better represent the findings in the laboratory setting. We have toned down our statement regarding the relevance of the results for ZIKV epidemiology (Line 126-128, Page 7; Line 254-257, Page 11; Line 294-296, Page 12; Line 306-307, Page 13; Line 325-331, Page 13; Line 337-339, Page 14; Line 364-375, Page 15). We also emphasize the potential limitations of this study (Line 315-320, Page 13; Line 350-359, Page 14).

#1B. *These concerns are not meant to prevent publication of the data, which are interesting. I would simply urge the authors to change their title, highlight the limitations of the study a bit more in the discussion, and instead highlight the main conclusion that can be drawn – that low levels of infectious Zika virus present in water during can infect mosquitoes during development and result in mosquitoes capable of transmission.*

#1B Answer: We appreciate the reviewer's positive comment for publishing the data. According to the suggestions, we have changed the title to "Zika viraemia causes infection in mosquito by breeding in infectious aquatic environments" to better represent the findings in the laboratory settings. We have toned down our statement regarding relevance of the results for ZIKV epidemiology (Line 126-128, Page 7; Line 254-257, Page 11; Line 294-296, Page 12; Line 306-307, Page 13; Line 325-331, Page 13; Line 337-339, Page 14; Line 364-375, Page 15). We also emphasize the potential limitations of this study (Line 315-320, Page 13; Line 350-359, Page 14). The current conclusion in the revised manuscript has been concentrated on the observations under the laboratory settings.

#1C. *Overall, there are still numerous English language errors and grammar issues throughout the manuscript. I urge the authors to proof-read and correct these. In some cases these affect clarity, e.g. lines 802-803 'The data for the upper mosquito*

number are represented as the percentage of infection' – this sentence makes little sense to me. Other grammar errors occur but are not limited to lines 101, 104, 121, and others including figure legends.

#1C Answer: Thank you for the reviewer's comments. The sentence in lines 802-803 has been replaced by "The percentages are represented as the ratios of mosquito infection." (Line 697, Page 29; Line 720, Page 30; Line 757, Page 31; Line 799, Page 33). We have comprehensively revised the manuscript by using an English language editing service. The errors noted by the reviewer have been corrected (Line 98, Page 6; Line 101, Page 6; Line 118, Page 6).

#1D. *It is not always clear what the 'mock' control is that the authors are showing. In Figure 3d for example, was the mock control just incubated in water? Or kept in urine? Similarly, the authors should check all figure legends and explain what the mock controls represent, where necessary.*

#1D Answer: We have clarified the "mock" controls in the figure legends (Line 684, Page 29; Line 733, Page 30; Line 739, Page 31).

#1E. *The authors mention in response #1B that 'the infectious supernatant was further subjected to viral sequencing to confirm that the cytopathic effects are caused by ZIKV infection'. However, I did not find this information in the methods or main text. I think it would be valuable to add that this was done to the manuscript since it is a concern when using unclean samples such as these for plaque assay (e.g. adding it to the methods after line 476).*

#1E Answer: We have added the experimental procedure in the Methods (Line 443-447, Page 18).

#1F. *Consistent use of a log scale on y-axes would be preferred to switching back and forth (e.g. Figure 3h-j).*

#1F Answer: We have made the changes in the figures according to the reviewer's suggestion (Figure 2f; Figure 3i).

#1G. *Line 356-359: 'with the accumulating ZIKV infectious particles discharged by a large number of patients'. The authors should keep in mind that for each infected patient that may have infectious ZIKV in their urine, there are probably multiple uninfected people (and patients without infectious ZIKV in their urine) discharging urine that would dilute the virus. This statement thus seems a bit grand and not entirely warranted.*

#1G Answer: Based on the suggestion of the reviewer, we have toned down the claims and revised the sentence to "We therefore speculate a scenario during a Zika outbreak in which the accumulating ZIKV infectious particles discharged by patients might be sufficient to facilitate the acquisition of an infection by immature mosquitoes breeding in cesspit conditions, thereby establishing the "host urination-mosquito breeding" transmission cycle." (Line 327-331, Page 13).

Responses to Referee #2:

#2A. *The authors have done an impressive amount of additional work to attempt to address the reviewer concerns. In particular, the use of sewage and lower viral concentrations are important additions. However, I am still far from convinced as to the epidemiological relevance of this observation. The additional experiments have highlighted that ZIKV doesn't survive very long in realistic conditions (Figure 3b). Importantly, readers are likely to be focused on the finding in the abstract that 5-17% of infections at the peak of the epidemics in French Polynesia came from urine infections – however, the evidence for this is very weak. I would strongly urge the authors to concentrate the article on the observation that under certain conditions, ZIKV can be transmitted from urine to mosquitoes and not make broad conclusions of the epidemiological relevance.*

#2A Answer: We agree with the reviewer's concern. According to the suggestions, we have changed the title to "Zika viruria causes infection in mosquito by breeding in infectious aquatic environments" to better represent the findings in the laboratory setting. We have toned down our statement regarding the relevance of the results for ZIKV epidemiology (Line 126-128, Page 7; Line 254-257, Page 11; Line 294-296, Page 12; Line 306-307, Page 13; Line 325-331, Page 13; Line 337-339, Page 14; Line 364-375, Page 15). We also emphasize the potential limitations of this study (Line 315-320, Page 13; Line 350-359, Page 14). The current conclusion in the revised manuscript has been concentrated on the observations under laboratory conditions.

#2B. *The major problem with the model is with the epsilon estimate. Epsilon represents the critical probability of ZIKV transmission during mosquito breeding in contaminated water. In Moorea, the authors provide an estimate of 0.49 with a standard deviation of 0.24, which will represent 95% confidence intervals of the entire range of probabilities (i.e. 0 to 1) – i.e., there is no information. The estimates in Tahiti are similar. In their own experiments, the authors are only able to infect a tiny proportion of mosquitoes in carefully selected conditions (i.e., only in a subset of sewage samples and urine samples could they get any infections) – therefore a probability of infection of 50% makes no sense. This parameter determines the transitions from the sewage compartment to the mosquito compartment so the overall proportion of infections that come from sewage is completely sensitive to this parameter.*

#2B Answer: Based on the reviewer's concerns and editorial suggestion, we decided to remove the modeling section from the revised manuscript.

#2C. *In the recent experiments, the authors use sewage samples. They previously demonstrated that in water, ZIKV survived up to 192 hours. With the sewage samples, they appeared to have stopped the experiments after 8 hours (at which point the pfu had dropped a lot and was only detectable at all in 4 samples, Figure 3b). Even if it shows a negative result – the authors should present a longer time-frame of the*

survivability in these more realistic conditions.

#2C Answer: According to the reviewer's suggestions, we have repeated the experiments and prolonged the time-frame of detection to 24 hrs (Figure 3b).

#2D. *Relatedly, in the new experiments, the authors also stopped the experiments after 24 hours when looking at survivability in urine at 28C (at which point it was only detectable in a single sample, Figure 1b) – again even if it shows a negative result, the authors should extend this analysis so that it is consistent with that in Fig 1a.*

#2D Answer: According to the reviewer's suggestions, we have repeated the experiments and prolonged the time-frame of detection to 48 hrs (Figure 1b).

#2E. *28C is also a high water temperature to be using as a realistic temperature for urine in sewage systems, even in tropical settings where ZIKV has been observed. In particular, nighttime temperatures will be important and likely to be substantially lower. At the very least, this weakness should be discussed.*

#2E Answer: We agree with the reviewer's concern. Indeed, the ZIKV epidemic area in Brazil located in tropical regions, where the maximum daytime temperature is 28.6°C-32.8°C and the minimum temperature ranges from 20.5°C -22.3°C at night (Alvares et al., 2013, Theor Appl. Climatol., 113, 407-427). However, the mosquitoes maintained in the laboratory are routinely reared at 28°C as a standard condition (Pompon et al., 2017, Sci. Rep. 7, 1215; Liu et al., 2017, Nature, 545,482-486; Mem. Inst. Oswaldo Cruz. 2017, 112, 829-837). Taking all of these studies into account, we selected 28°C as a temperature parameter in our laboratory settings to mimic the natural conditions. We noted that the 28°C temperature might represent a realistic daytime condition in the ZIKV epidemic area. Nonetheless, nighttime temperatures are also essential for mosquito breeding and are likely to be substantially lower. Therefore, both mosquito breeding and the ZIKV stability under various temperature conditions remain to be further investigated. We have added this to the discussion (Line 350-359, Page 14)

#2F. *There remain a lot of grammatical errors – I found the paper difficult to follow in a number of places.*

#2F Answer: We have comprehensively revised the manuscript by an English language editing service.

Responses to Referee #3:

#3A. *Similar to Reviewer 1, I believe that the concept that mosquitoes can acquire ZIKV from sewage water is interesting and worthy of publication on its own without the grandiose conclusions about contributions to transmission. It is fine to speculate about some of this but in so doing it is necessary for the authors to discuss potential limitations to their experimental design and acknowledge the need for epidemiological follow up.*

#3A Answer: We agree with the reviewer's concern. According to the suggestions, we have changed the title to "Zika viremia causes infection in mosquito by breeding in infectious aquatic environments" to better represent the findings in the laboratory settings. We have toned down our statement regarding the relevance of the results for ZIKV epidemiology (Line 126-128, Page 7; Line 254-257, Page 11; Line 294-296, Page 12; Line 306-307, Page 13; Line 325-331, Page 13; Line 337-339, Page 14; Line 364-375, Page 15). We also emphasize the potential limitations of this study (Line 315-320, Page 13; Line 350-359, Page 14). The current conclusion in the revised manuscript has been concentrated on the observations under laboratory conditions.

#3B. *For Line 197-201, I agree that low viremia has been reported in the literature but I disagree with the speculation that mosquitoes may have a low overall vector competence for ZIKV based on these data and the author's demonstration of poor infection rates for mosquitoes exposed to bloodmeal titers <10E5 PFU/ml. 1.) Viremia titers reported in the literature from infected humans are difficult to interpret because they are a snapshot in time and in all likelihood the result of an individual seeking treatment after symptom onset. As a result, it is possible that these titers may be past peak titer in that individual. 2.) Membrane feeding can influence the magnitude of the observed effect as compared to feeding on a viremic host, i.e., mosquitoes that are allowed to feed on viremic hosts (in general) have higher infection, dissemination, and transmission rates as compared to mosquitoes exposed to comparable bloodmeal titers via membrane feeding systems.*

#3B Answer: We agree with the reviewer's concerns. Accordingly, we have modified the statement in the Results section (Line 195, Page 9). In the Discussion, we have also emphasized that the human viremic titers represented in the previous literature may reflect snapshot results of patients seeking treatment after symptom onset. Therefore, it is possible that these titers may reflect off-peak viremia. (Line 339-341, Page 14).

#3C. *I still continue to question the relevance of the mouse transmission model utilized here. At the very least there should be some discussion about the overall susceptibility of this model to Zika virus infection as I indicated in my previous review. This may help reconcile the very low infection rate detected when mosquito pools are screened vs. the low-to-moderate transmission rate to mice.*

#3C Answer: In the first experiment with the Rockefeller strain and human urine, we tested the ZIKV infectivity in mosquitoes breeding in ZIKV urine containing 20 pfu/ml ZIKV. Two of 50 Rockefeller *A. aegypti* pools (10 mosquitoes per pool) were positive for infectious ZIKV (Figure 1f). The ratios of mosquito infection may range from **0.4% (2/500)-4% (20/500)**. In the correlated transmission experiment, twenty female *A. aegypti* mosquitoes, which bred from 20 pfu/ml ZIKV urine, were allowed to feed on an AG6 mouse. Three out of the 17 mice exposed to infected mosquitoes were ZIKV-positive (Figure 2b). In this transmission study, the estimated ratios of mosquitoes with the capacity for viral transmission may range from **0.88%**

(3/340)-17.6% (60/340).

In the other parameter settings of the field-derived Brazil mosquitoes and sewage conditions, one out of 78 mosquito pools (10 mosquitoes per pool) from the sewage with 1/10 dilution of 20 pfu/ml ZIKV urine was positive for the isolation of infectious ZIKV in Vero cells (Figure 3f). The ratios of mosquito infection may range from **0.13% (1/780)-1.3% (10/780)**. In the transmission study, thirty female *A. aegypti* mosquitoes were allowed to feed on an AG6 mouse. One out of the 18 mice exposed to infected mosquitoes was ZIKV-positive by either RT-qPCR (Figure 3h) or plaque assay (Figure 3i) after mosquito feeding, suggesting that the estimated ratios of mosquitoes with capacity for viral transmission may range from **0.19% (1/540)-1.9% (10/540)**.

Based on the infective mosquito ratios in the 2 independent experiments, we concluded that the infective ratios detected in mosquito pools correlate well with the transmission rate to mice.

#3D. *I still question the veracity of the model and am concerned about potential misinterpretation.*

#3D Answer: Based on the reviewer's concerns and editorial suggestion, we decided to remove the modeling section from the revised manuscript.